# Spatiotemporal dynamics of clonal selection and diversification in normal endometrial epithelium

Manako Yamaguchi [1,9], Hirofumi Nakaoka [2,3,9✉], Kazuaki Suda[1,9], Kosuke Yoshihara [1,9✉], Tatsuya Ishiguro [1], Nozomi Yachida[1], Kyota Saito[1], Haruka Ueda[1], Kentaro Sugino[1], Yutaro Mori[1], Kaoru Yamawaki[1], Ryo Tamura[1], Sundaramoorthy Revathidevi[2], Teiichi Motoyama[4], Kazuki Tainaka[5,6], Roel G. W. Verhaak [7,8], Ituro Inoue[2✉] & Takayuki Enomoto [1✉]

It has become evident that somatic mutations in cancer-associated genes accumulate in the normal endometrium, but spatiotemporal understanding of the evolution and expansion of mutant clones is limited. To elucidate the timing and mechanism of the clonal expansion of somatic mutations in cancer-associated genes in the normal endometrium, we sequence 1311 endometrial glands from 37 women. By collecting endometrial glands from different parts of the endometrium, we show that multiple glands with the same somatic mutations occupy substantial areas of the endometrium. We demonstrate that "rhizome structures", in which the basal glands run horizontally along the muscular layer and multiple vertical glands rise from the basal gland, originate from the same ancestral clone. Moreover, mutant clones detected in the vertical glands diversify by acquiring additional mutations. These results suggest that clonal expansions through the rhizome structures are involved in the mechanism by which mutant clones extend their territories. Furthermore, we show clonal expansions and copy neutral loss-of-heterozygosity events occur early in life, suggesting such events can be tolerated many years in the normal endometrium. Our results of the evolutionary dynamics of mutant clones in the human endometrium will lead to a better understanding of the mechanisms of endometrial regeneration during the menstrual cycle and the development of therapies for the prevention and treatment of endometrium-related diseases.

[1] Department of Obstetrics and Gynecology, Niigata University Graduate School of Medical and Dental Sciences, Niigata 951-8510, Japan. [2] Human Genetics Laboratory, National Institute of Genetics, Mishima 411-8540, Japan. [3] Department of Cancer Genome Research, Sasaki Institute, Sasaki Foundation, Chiyoda-ku 101-0062, Japan. [4] Department of Molecular and Diagnostic Pathology, Niigata University Graduate School of Medical and Dental Sciences, Niigata 951-8510, Japan. [5] Department of System Pathology for Neurological Disorders, Brain Research Institute, Niigata University, Niigata 951-8585, Japan. [6] Laboratory for Synthetic Biology, RIKEN Center for Biosystems Dynamics Research, Suita 565-5241, Japan. [7] The Jackson Laboratory for Genomic Medicine, Farmington, CT, USA. [8] Department of Neurosurgery, Cancer Center Amsterdam, Amsterdam UMC, VU University Medical Center (VUmc), 1081 HV Amsterdam, The Netherlands. [9]These authors contributed equally: Manako Yamaguchi, Hirofumi Nakaoka, Kazuaki Suda, Kosuke Yoshihara. ✉email: hirofumi.nakaoka.5474@gmail.com; yoshikou@med.niigata-u.ac.jp; itinoue@nig.ac.jp; enomoto@med.niigata-u.ac.jp

Cancer is a collection of diseases characterized by uncontrollable cellular growth and spread caused by somatic mutations[1,2]. Genes whose mutations confer selective growth advantages on cells are called cancer drivers or cancer-associated genes[3]. With the advent of next-generation sequencing technologies, large-scale cancer genome projects have identified hundreds of cancer-associated genes[4,5]. The lifetime incidence of cancer in different tissues is correlated with the number of stem cell divisions in the corresponding tissues, suggesting that cancer-associated gene mutations randomly arise and accumulate in human adult stem cells during life[6,7]. Recent studies have demonstrated that cancer-associated gene mutations lurk not only in benign tumors but also in histologically normal tissues[8] such as bladder[9], blood[10], colon[11–13], endometrium[14–17], esophagus[18–20], liver[21], lung[22], skin[23], and ureter[24]. Cells carrying cancer-associated gene mutations are thought to manifest increased cellular fitness over their neighboring cells and to be positively selected within normal tissues[25]. However, it remains to be elucidated when and how mutant clones expand their territories in normal tissue. It is crucial to investigate the distribution of cancer-associated gene mutations across time and space[26].

Apart from higher primates, menstruation is quite rare in the animal kingdom[27]. The human endometrium has a unique capability to cyclically regenerate and remodel throughout a woman's reproductive life. Histologically, the human endometrium is stratified into the stratum functionalis and the stratum basalis. The functionalis is eroded during menstruation and regenerates from the remaining basalis[28]. The highly regenerative nature of this tissue poses risks for developing endometrium-related diseases such as adenomyosis, endometrial hyperplasia, endometriosis, and endometrial and ovarian cancer in adult women. The normal endometrium can be a source of mutant clones that lead to the development of endometrium-related diseases. By sequencing histologically normal endometrial glands, we and others have identified numerous somatic mutations in cancer-associated genes, such as PIK3CA and KRAS[15–17]. Individual glands carry distinct somatic mutations in clonal states, verifying that the composition of each gland is monoclonal[29] but that gland-to-gland variation shapes the mosaic-like genomic composition of the uterine endometrial epithelium[15,17]. Additionally, we found that several glands had identical mutation profiles, even though we randomly collected glands by enzymatically dissociating nearly the entire surgically resected part of the endometrium, implying the presence of mutant clones occupying certain areas of histologically normal endometrium[15].

The latest three-dimensional (3D) imaging analyses have revealed that the morphology of the human endometrium is much more complicated than previously believed[30,31]. In particular, plexus network structures of endometrial glands within the basalis were discovered in humans[30,31] but not in mice[32,33]. Mutation profiling of endometrial glands guided by 3D imaging techniques can help illuminate mechanisms by which somatic mutations spread within the endometrial epithelium.

In this work, we perform target-gene sequencing, whole-exome sequencing (WES), and whole-genome sequencing (WGS) for 1311 endometrial glands from 37 women across a wide range of ages. Combined with the estimation of chronological ages at which genome events such as clonal expansions and somatic copy number alterations occurred, our sequencing analyses preserving the spatial information of endometrial glands provide fundamental clues to understand the spatiotemporal dynamics of clones with cancer-associated gene mutations in the normal endometrium.

## Results

**Cancer-associated gene mutations in normal endometrial glands.** We conducted target-gene sequencing of 112 genes in 891

normal endometrial glands from the uteri of 32 subjects ranging in age from 21 to 53 years old (Fig. 1a, Supplementary Table 1 and Supplementary Data). More than half of the normal endometrial glands acquired numerous somatic mutations in genes that are frequently mutated in endometrial cancer and endometriosis-associated ovarian cancer (Fig. 1b and Supplementary Fig. 1). The most frequently mutated genes included PIK3CA (15.6%), KRAS (10.9%), FBXW7 (8.1%), PIK3R1 (7.1%), and PPP2R1A (6.7%) (Fig. 1c). The mutant allele frequencies (MAFs) of these mutations were close to 0.5, indicating their clonal expansion within the glands (Fig. 1d). Moreover, the MAFs of the mutations exhibited a trimodal distribution presumably composed of subclonal mutations (<0.25), clonal mutations (≥0.25), and clonal mutations with allelic imbalance, such as subsequent loss of the wild-type allele (≥0.75) (Fig. 1e). 61.8% (=551/891) and 50.1% (=446/891) of glands harbored high and low MAF SNVs, respectively. The presence of both clonal and subclonal mutations suggests that a cell clonally expands to form an endometrial gland, and the clone then acquires additional diversifying mutations. KRAS mutations in 15 endometrial glands exhibited allelic imbalance favoring mutant alleles (MAF ≥0.8, binomial test $P < 10^{-5}$).

**Somatic mutations accumulate with age and cumulative number of menstrual cycles (CNMCs).** To dissect active mutational processes, we classified the somatic single-nucleotide variants (SNVs) with high MAF (≥0.25) into 96 mutation classes constituted by the six pyrimidine substitutions (C>T, C>A, C>G, T>C, T>G, and T>A) in combination with the flanking 5′ and 3′ bases (Fig. 1f). We explored mutational signatures characterizing the mutational processes operative in normal endometrial glands, where the spectrum of somatic SNVs with high MAF was fitted to a set of the Catalogue of Somatic Mutations in Cancer (COSMIC) mutational signatures[34] ("Methods"). Three mutational signatures (SBS1, SBS5, and SBS18) were significantly overrepresented (Fig. 1g). SBS1 is a clock-like mutation signature characterized by C>T transitions at CpG motifs, indicative of the deamination of methyl-cytosines[35]. SBS5 is another clock-like mutation signature with unknown etiology[36]. SBS18, represented by C>A transversions, has been attributed to DNA damage induced by reactive oxygen species[37].

We calculated the burden of somatic mutations for each subject over glands. The burden varied substantially among the subjects (Fig. 2a and Supplementary Fig. 1). We examined whether the burden of somatic mutations was associated with the clinical characteristics of the subjects. The burden of somatic SNVs with high MAF showed strong linear relationships with age (Pearson's correlation coefficient, $r = 0.79$; $P = 7.5 \times 10^{-8}$) and CNMCs ($r = 0.81$; $P = 2.4 \times 10^{-8}$) (Fig. 2b). CNMCs explained the burden of somatic SNVs with high MAF better than age, probably because menarche was negatively associated with the burden after adjustment for age ($r = -0.30$; $P = 0.09$) (Fig. 2c, Supplementary Fig. 2a). Moreover, both age and CNMCs were significantly associated with the burden of C>T transitions at CpG motifs (Fig. 2d and Supplementary Fig. 2d) and with the burdens of C>T, C>A, C>G, and T>C substitutions, but not with the burden of T>G or T>A substitutions (Fig. 2e and Supplementary Fig. 2e), reminiscent of the abovementioned two mutational signatures with clock-like properties (SBS1 and SBS5).

Next, we examined the burden of driver mutations that were defined as non-silent SNVs and short insertions and deletions (indels) with high MAF in cancer-driver genes (ARID1A, CTNNB1, FBXW7, KRAS, PIK3CA, PIK3R1, PPP2R1A, PTEN, and TP53). The burden of driver mutations was significantly associated with age ($r = 0.63$; $P = 9.9 \times 10^{-5}$) and CNMCs

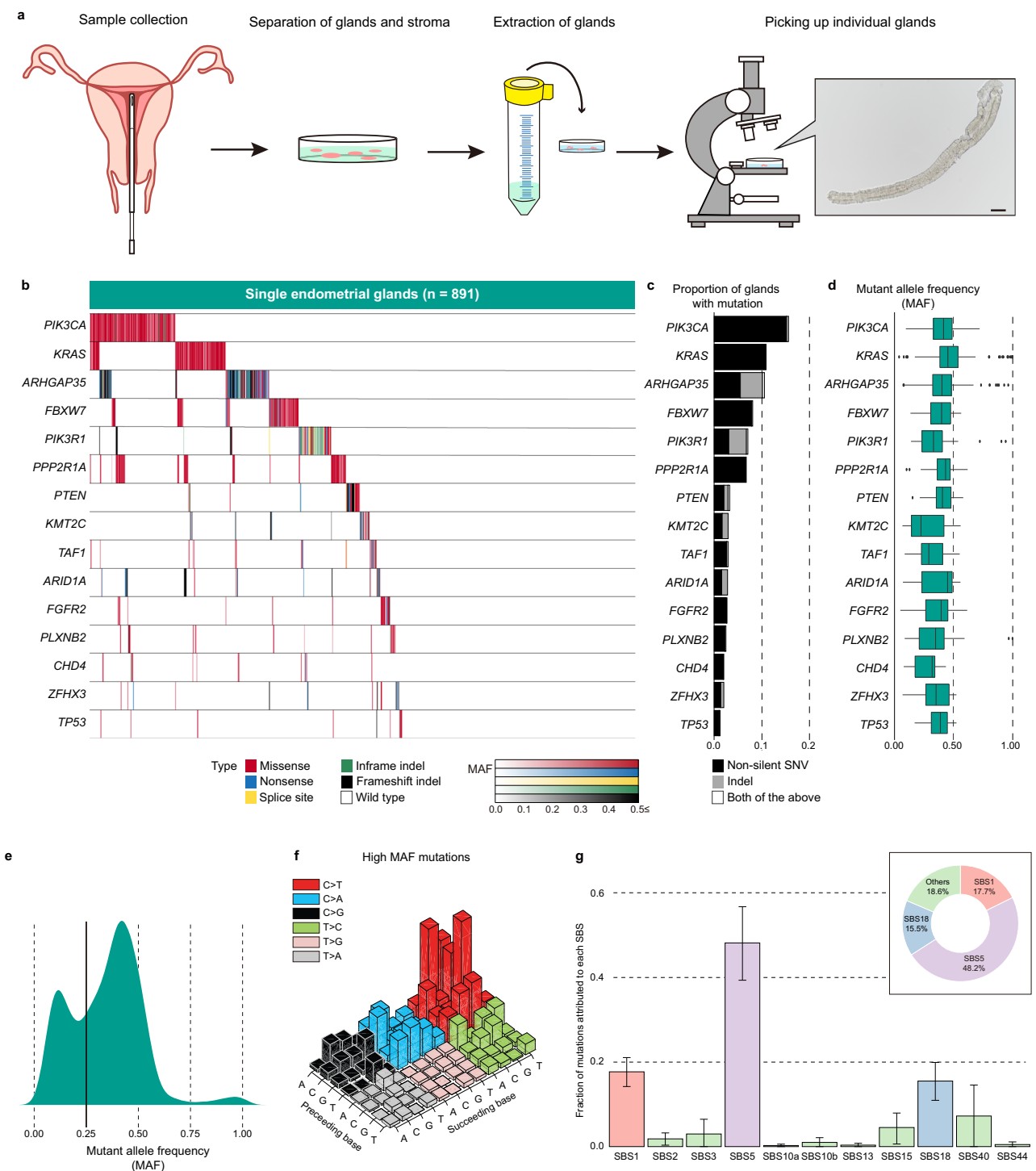

(r = 0.69; P = 1.3 × 10⁻⁵) (Supplementary Fig. 3). Again, CNMCs explained the burden of driver mutations better than age, because age of menarche showed a significant negative correlation after the adjustment for age (r = −0.50; P = 3.4 × 10⁻³). The effects of parity, body mass index (BMI), pack years of smoking and disease status were not significantly associated with the burdens of somatic SNVs with high MAF and driver mutations after the adjustment for age (Supplementary Figs. 2b, c and 3).

The average age and CNMCs of patients with *KRAS* mutations exhibiting allelic imbalance (MAF ≥ 0.8) were higher (P = 0.041 and P = 0.021, respectively; Wilcoxon rank-sum test). Additionally, mutations exhibiting allelic imbalance were overrepresented

in patients with endometriosis (P = 0.012; Fisher's exact test). Further studies are needed to validate the associations between *KRAS* mutations exhibiting allelic imbalance and clinical variables.

**Strong positive selection acting on cancer-associated genes.** The distributions of the cancer-associated gene mutations in normal endometrial glands were similar to those in cancers based on the COSMIC database[38] (Fig. 3a–f and Supplementary Fig. 4). Additionally, nonsilent mutations predominated over silent mutations in the cancer-associated genes. Therefore, we assessed

**Fig. 1 Landscape of somatic mutations in normal endometrial glands. a** An outline flowchart of isolation of single endometrial glands. First, the samples were collected from endometrial biopsies using suction catheters or endometrial curettage. Second, endometrial tissues digested with collagenase to separate glands from stroma. Third, the solution of collagenase-digested tissue was poured into a cell strainer. The epithelial cell glands remaining on the strainer were transferred into a cell culture dish. Finally, individual glands were picked up precisely under a microscope. Scale bar in microscopic images, 100 μm. **b** Heatmap of the prevalence of nonsilent somatic mutations in 891 normal endometrial glands from the uteri of 32 subjects. Nonsilent mutations are color-coded as follows: missense SNVs (red), nonsense SNVs (blue), splice-site SNVs (yellow), in-frame indels (green) and frameshifting indels (black). Color density indicates the MAF of each somatic mutation. **c** Bar charts representing proportions of endometrial glands with somatic mutations in 15 cancer-associated genes stacked by mutation type: SNVs (black), indels (gray), and both (white). **d** Box-whisker plots showing distributions of the MAFs of somatic mutations on the 15 genes. The number of mutations in each of the 15 genes (*n*), from top to bottom, is 139 (*PIK3CA*), 97 (*KRAS*), 98 (*ARHGAP35*), 72 (*FBXW7*), 66 (*PIK3R1*), 60 (*PPP2R1A*), 32 (*PTEN*), 26 (*KMT2C*), 26 (*TAF1*), 25 (*ARID1A*), 24 (*FGFR2*), 24 (*PLXNB2*), 18 (*CHD4*), 18 (*ZFHX3*), and 11 (*TP53*). Box plots show the minima (leftmost dot), the maxima (rightmost dot), the median (middle line) and the first and third quartiles (boxes), whereas the whiskers show 1.5× the interquartile range (IQR). **e** Density plot showing trimodal distributions of the MAFs of all the identified somatic mutations. The vertical line at an MAF of 0.25 represents the threshold for dichotomizing mutations into low- and high-MAF mutations. **f** Lego plot depicting the mutation spectrum of somatic SNVs with high MAF in normal endometrial glands. **g** Bar chart representing the results of Bayesian inference by using sigfit to determine the contribution of the COSMIC mutational signatures to somatic SNVs with high MAF. Data are presented as the estimated contributions of significant mutational signatures and the lower and upper limits of the 90% highest posterior density interval. The inset is a doughnut chart summarizing the contributions of three significant mutational signatures (SBS1, SBS5, and SBS18). The number of high MAF SNVs (*n*) used for the analysis is 810. Source data are provided as a Source data file.

whether the somatic mutations in cancer-associated genes were under positive and negative selection in normal endometrial glands based on the dN/dS ratio[39] ("Methods"). First, we evaluated the extent of selection acting on a set of genes. The dN/dS ratios of missense mutations for all 112 targeted genes, 48 genes in the Cancer Gene Census[40], and 15 genes in the pan-gynecologic cancer-associated genes[41] were 3.13 (95% confidence interval [CI], 2.38-4.12; $P = 4.4 \times 10^{-16}$), 4.92 (95% CI, 3.38–7.16; $P < 10^{-16}$), and 14.5 (95% CI, 7.20–29.2; $P = 7.1 \times 10^{-14}$), respectively (Fig. 3g). The dN/dS ratios of nonsense mutations for these sets of genes were 6.15 (95% CI, 4.35–8.68; $P < 10^{-16}$), 5.88 (95% CI, 3.63–9.53; $P = 6.7 \times 10^{-13}$) and 11.1 (95% CI, 4.95–24.7; $P = 4.7 \times 10^{-9}$), respectively (Fig. 3g). Notably, the extent of positive selection was strongest in the pan-gynecologic cancer-associated genes. According to the dN/dS ratios, up to 93.1% of the missense and 91.0% of the nonsense mutations in the pan-gynecologic cancer-associated genes were estimated to be drivers with strong selective advantages in the normal endometrium. On the contrary, the dN/dS ratios of missense and nonsense mutations in genes other than cancer-associated genes were 0.98 (95% CI, 0.63–1.51; $P = 0.91$) and 1.20 (95% CI, 0.52–2.75; $P = 0.67$), respectively (Fig. 3g). This result suggests that the mutations in the non-cancer-associated genes were selectively neutral (passengers).

Next, we explored signals of selection at the level of individual genes. Missense mutations of *KRAS*, *PIK3CA*, *PPP2R1A*, *FBXW7*, *PTEN*, *PIK3R1*, *TP53*, *FGFR2*, *ARHGAP35*, and *PLXNB2* exhibited strong signals of positive selection (false discovery rate (FDR) < 0.1; Benjamini–Hochberg procedure) (Fig. 3h). In addition, nonsense substitution rates in *ARHGAP35*, *PTEN*, *PIK3R1*, *FBXW7*, *ARID1A*, *ARID5B*, *TAF1*, *ZFHX3*, and *KMT2C* significantly exceeded synonymous substitution rates (FDR < 0.1) (Fig. 3h). Remarkably, two representative oncogenes, *KRAS* and *PIK3CA*, showed biased signs of positive selection toward the excess of missense substitution rates without observing nonsense substitutions (Fig. 3i). Tumor suppressor genes presented several patterns: *PTEN*, *FBXW7*, and *PIK3R1* showed excesses of both missense and nonsense substitution rates. *ARHGAP35* and *ARID1A* showed biased signs toward excess nonsense substitution rates, while missense substitutions were predominant in *PPP2R1A* and *TP53* (Fig. 3i).

Although we randomly collected endometrial glands from the entire uterus, 15 out of 32 subjects had at least one pair of glands that identically shared multiple high-MAF mutations (Supplementary Fig. 5). Glands from two subjects aged 47 showed higher

burdens of identically shared multiple high-MAF mutations. Seven out of 27 glands from Subject 27 harbored identical *KRAS* (p. G12C) and *PIK3CA* (p. G118D) mutations. Four out of 16 glands from Subject 28 shared *KRAS* (p. G12D) and *SPEG* (p. A2522=) mutations. These findings imply that glands with the same clonal origin spread over a wide area of the endometrium, particularly in aged women.

**Spatially resolved single endometrial gland sequencing**. We conducted target-gene sequencing for spatially resolved single endometrial glands, in which surgically resected specimens of normal endometrium from four subjects were subdivided into grids, and endometrial glands were collected from the grids. Thus, spatial information of each gland was retained at the grid level (Figs. 4 and 5, Supplementary Fig. 6 and Supplementary Table 2). We sought informative mutations that were shared among multiple endometrial glands. Then, we performed hierarchical clustering of the MAF profiles of the informative mutations.

We divided the endometrium from a 38-year-old subject with endometriosis (Subject 33) into 24 grids with 3.5 mm squares and extracted three glands from each grid (Fig. 4a). We identified two clusters characterized by *PIK3CA* (p.E545A) and *KIF26A* (p.P326L) and by *KRAS* (p.G12S) and *PIK3CA* (p.E542V). The glands in each cluster were located within the same grid (Fig. 4b, c).

For a subject aged 41 years with myoma uteri (Subject 34), we examined five glands from each of 24 grids with 5 mm squares (Fig. 4d). We detected four clusters of glands with distinct mutation profiles (Fig. 4e, f). The largest cluster characterized by two mutations (*ZFHX4* [p. A527V] and *PPP2R1A* [p.S219L]) occupied five adjacent grids (3, 11, 12, 14, and 15), spanning ~2 cm in the largest diameter. Glands carrying only *ZFHX4* (p.A527V) were present in the central region (grids 7 and 12) linking the abovementioned five grids. We further investigated the clonal origin of these glands with a larger number of somatic mutations by WES. The glands with only *ZFHX4* (p.A527V) shared additional somatic mutations with the glands harboring both *ZFHX4* (p.A527V) and *PPP2R1A* (p.S219L) (Supplementary Fig. 6e). The phylogenetic tree obtained by WES suggests that the glands in these two groups have a common ancestral clone harboring *ZFHX4* (p.A527V) from which two descendant clones diversified, one of which acquired *PPP2R1A* (p.S219L) and then spread to the broad area (Fig. 4g).

The anterior and posterior walls of the endometrium from a 46-year-old subject with cervical carcinoma in situ (Subject 35)

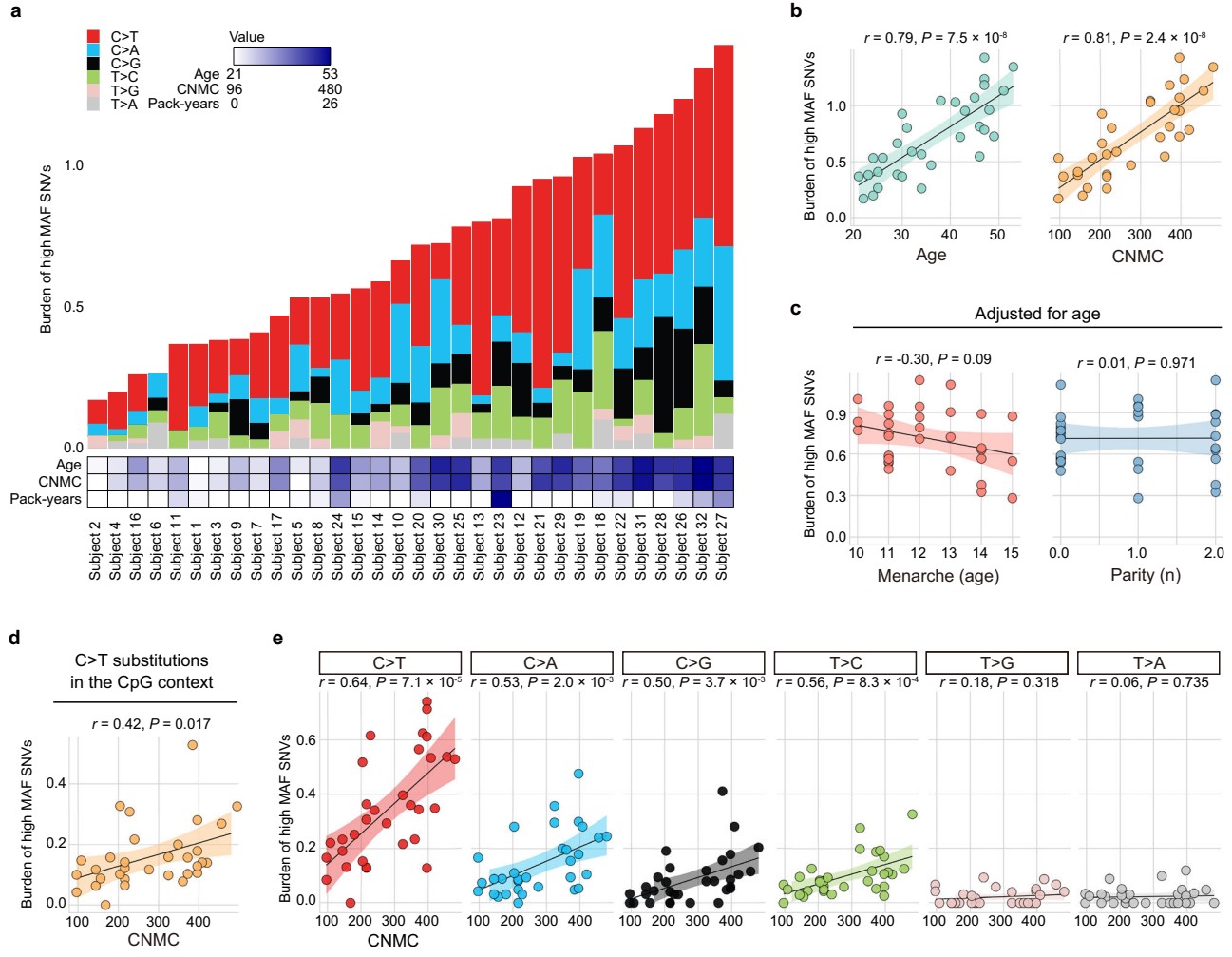

**Fig. 2 Burden of somatic mutations accumulates with age and CNMCs. a** Bar charts showing the burdens of somatic SNVs with high MAF for 32 subjects stacked by the six pyrimidine substitutions (C>T, C>A, C>G, T>C, T>G, and T>A). The burden is defined as the number of high-MAF SNVs per Mbp sequenced. Multiple glands from an individual subject are pooled. Subjects are sorted in ascending order according to mutation burden. The heatmap below the bar charts represents age, CNMCs, and pack-years of cigarette smoking for the 32 subjects. **b** Linear relationships of age and CNMCs with the burden of somatic SNVs with high MAF. **c** Linear relationships of age of menarche and number of parities with the burden of somatic SNVs with high MAF after adjustment for age. **d** Linear relationship of CNMCs with the burden of C>T transitions at CpG motifs. **e** Linear relationships of CNMCs with the burdens of the six pyrimidine substitutions. **b–e** Pearson's correlation coefficient (r) and the statistical significance level (P) based on two-tailed test are shown. Unadjusted P-value is shown. The slope of a linear regression line (solid line) with 95% CI (shaded area as error band) is plotted. Source data are provided as a Source data file.

were resected. Each of them was divided into 24 grids with 4 mm squares (Fig. 4h). The sequencing of three glands from each grid showed six and nine clusters with distinct mutations in the anterior and posterior walls, respectively (Fig. 4i–k). Glands in nine clusters were distributed among multiple neighboring grids, suggesting a high level of mosaicism in the endometrium of this subject. Glands in six clusters had the same mutation of *FGFR2* (p.S252W). However, WES showed that the glands in different clusters did not share any mutations other than *FGFR2* (p.S252W) (Supplementary Fig. 6f). Additionally, glands in these six clusters were localized in spatially separated regions of the endometrium: Cluster 3 was distributed in the center of the anterior wall, cluster 12 was in the upper right of the posterior wall, and clusters 8, 13–15 were in the lower left of the posterior wall (Fig. 4i). It is not likely that the *FGFR2* mutation initially occurred in a single ancestral clone and the ancestral clone expanded to these three separated regions as depicted by the phylogenetic tree (Fig. 4l), because glands with the *FGFR2* mutation were not detected in regions connecting the three

regions. Therefore, a plausible explanation may be that at least three mutational events of the *FGFR2* mutation occurred independently in the three separated regions of the endometrium and the mutant clones diversified at each region by acquiring region-specific mutations.

More intensive screening was conducted in the endometrium from a 50-year-old subject with myoma uteri, where the 1 cm-square region was divided into four grids (A to D), and one of them (A) was further partitioned into four subgrids (A-I to A-IV) (Fig. 5a, Supplementary Fig. 6g–i). We extracted 13 or 15 glands from each of the subgrids and 20 glands from the remainder of the grids (B, C, and D). We identified five clusters: two were limited to within a grid or subgrid, and the other three were prevalent among multiple grids and subgrids (Fig. 5b, c). Two clusters (1 and 3) accounted for 18 glands spreading across three grids (B, C, and D) and one subgrid (A-IV) and 16 glands spreading across two grids (C and D), respectively. Cluster 3 had a frameshift insertion in *PTEN* with high MAF (>0.5), implying an allelic imbalance at *PTEN*, which is known to occur frequently

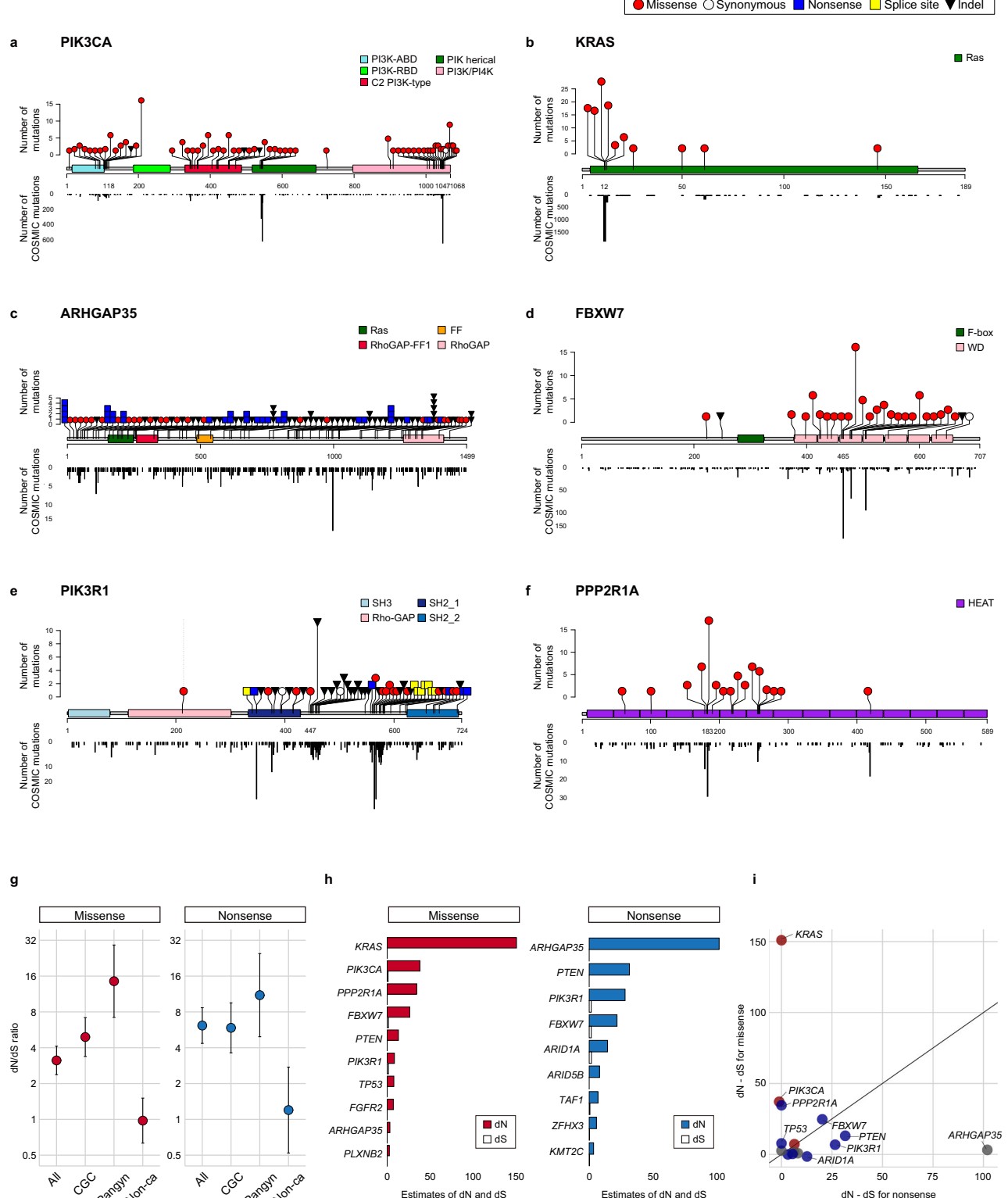

in endometrial carcinoma[42]. Using WGS, we confirmed that glands in the same clusters originated from common ancestral clones (Fig. 5d, e). Additionally, we identified two arm-level copy-neutral loss-of-heterozygosity (CN-LOH) events at chromosomes 3 and 10 in all glands of cluster 3 (Fig. 5f and Supplementary Fig. 7). As reflected in the branch lengths of the phylogenetic tree, the number of private mutations unique to single glands was larger in cluster 1 than in cluster 3 (Fig. 5d, e). We leveraged the

intracluster heterogeneity to infer the chronological ages at which genomic events emerged in the endometrium ("Methods"). The most recent clonal expansions of clusters 1 and 3 were estimated to have occurred at the ages of 38.3 (95% CI, 37.1–39.6) and 49.2 (95% CI, 48.9–49.5), respectively (Fig. 5g). This implies that stem/progenitor cells giving rise to glands of cluster 1 had spread their territories for more than 10 years and diverged by acquiring mutations, which led to local variability. In sharp contrast, the

**Fig. 3 Localization and positive selection of cancer-associated gene mutations in normal endometrial glands. a–f** Lollipop plots showing the locations of the identified somatic mutations in **a** PIK3CA, **b** KRAS, **c** ARHGAP35, **d** FBXW7, **e** PIK3R1, and **f** PPP2R1A along with known domain structures of the proteins encoded by these genes. Numbers refer to amino acid residues. The heights of the lollipops correspond to the number of mutations at each amino acid residue. Black bar charts indicate the number of somatic mutations deposited in the COSMIC database. **g** Estimated values of the dN/dS ratios for missense and nonsense mutations acting on a set of genes. The dN/dS ratios of missense mutations for all 112 targeted genes, 48 genes in the Cancer Gene Census (CGC), 15 genes in the pan-gynecologic cancer-associated genes (Pangyn) and non-cancer-associated genes. Non-cancer-associated genes were defined as a list of genes by excluding the CGC and Pangyn genes and ARHGAP35 from 112 targeted genes. The numbers (n) of missense, nonsense, and synonymous substitutions are 543, 76, and 56, respectively, in all 112 genes; 444, 38, and 29, respectively, in the CGC; 385, 23, and 8, respectively, in the Pangyn; and 81, 7, and 27, respectively, in the non-cancer-associated genes. Data are presented as the estimates of the dN/dS ratio and their 95% CIs. **h** Estimated values of dN and dS for missense and nonsense mutations at the level of individual genes. The dN and dS values are shown rather than the dN/dS ratios because the dS values for some genes were zero because of the absence of synonymous substitutions. Significant genes at FDR < 0.1 are shown. **i** Scatterplot of the estimated dN − dS values for missense mutations against those for nonsense mutations in 16 cancer-associated genes in (**h**). Oncogenes: red, tumor suppressor genes: blue, and other genes: gray. Source data are provided as a Source data file.

clone-producing glands of cluster 3 had undergone rapid growth within a short period. We investigated whether the CN-LOHs at chromosomes 3 and 10 contributed to the rapid expansion by examining the MAF profiles of public SNVs that were shared among all the glands in the cluster and located in the regions affected by the CN-LOHs (Fig. 5h). The rationale behind this analysis is that the MAFs of mutations that occurred before a CN-LOH are expected to become predominant, whereas mutations that occurred subsequently are expected to be half as prevalent because they are present in one of the two copies of the duplicated chromosome. Therefore, we can estimate the timing of the CN-LOH based on the proportion of public mutations with predominant status to the overall public mutations in the CN-LOH region[43,44]. Surprisingly, the burden of somatic SNVs that preceded the CN-LOH at chromosome 10 was lower than that of SNVs that occurred subsequently. On the other hand, the burden of somatic SNVs that preceded the CN-LOH at chromosome 3 exceeded that of those that occurred subsequently. Consequently, the CN-LOHs at chromosomes 10 and 3 were estimated to occur at the ages of 17.3 (95% CI, 11.7–22.8) and 35.5 (95% CI, 30.8–40.1), respectively (Fig. 5i, j), both decades earlier than the most recent clonal expansion.

**3D imaging of normal endometrial glands**. To elucidate the mechanisms by which endometrial glands with the same clonal origin expanded across spatially distant regions, we performed 3D imaging analysis on normal proliferative endometrial tissues from four middle-aged women who underwent hysterectomy due to gynecological diseases (Supplementary Table 3) by using a recently developed tissue clearing technique combined with light-sheet fluorescence microscopy[31,45]. By examining continuous tomographic images, we visualized the plexus structures linking a set of glands with their root at the basal layer (Fig. 6a, b and Supplementary Video 1). Specifically, the glandular structure at the bottom of the endometrium ran horizontally along the muscular layer, similar to a rhizome of grass (hereinafter called the "rhizome structure"), and several branches rose from the rhizome structure toward the luminal epithelium (Fig. 6c and Supplementary Video 1). The rhizome structures expanded linearly rather than radially. These properties were found in all four cases, suggesting that the presence of rhizome-sharing glands is common in middle-aged women (Supplementary Fig. 8).

To quantify the space occupied by a continuous set of rhizome structures and glands derived from the rhizome, we selected a representative rhizome from each patient. The number of rhizome-sharing glands ranged from 4 to 19 (Supplementary Fig. 8a–d). We determined the XYZ coordinates of the tips of rhizome-sharing glands to measure the longest distance between glands and the area occupied by the glands (Fig. 6d and Supplementary Fig. 8e–l). The rhizome-sharing glands reached

a length of 1.3–3.8 mm and occupied an area of 0.3–4.7 mm$^2$, indicating that they occupied a substantial part of the endometrium.

**WGS for selectively isolated endometrial glands based on 3D images**. To analyze the genomes of rhizome structures and the vertical glands derived from them, we examined 70 serial cryo-sections (12 µm thick) from the proliferative-phase endometrium of a 43-year-old woman who underwent hysterectomy for endometriosis (Fig. 7a–c). We detected two independent groups of glands that were in close proximity but separated from each other; the first group comprised rhizome G1 and vertical glands G2–G7, and the second group comprised rhizome G8 and vertical glands G9–G12 (Fig. 7b, c and Supplementary Fig. 9). Additionally, we found a single vertical gland (G13) that was independent of the two gland groups. Then, the 3D image was reconstructed to validate the spatial locations and continuity of the 13 glands (Fig. 7d and Supplementary Video 2). Guided by the 3D image analysis, we microdissected the 13 glands for WGS (Fig. 7a). The WGS showed that the first and second groups shared different sets of somatic mutations, but G13 did not share mutations with other glands, suggesting that each of the groups comprising a rhizome and vertical glands originated from a unique mutant clone (Fig. 7e). The profile for cancer-associated gene mutations also showed a group-specific segregation pattern (PIK3CA [p. E453K] in the first group, PTEN [p. Y188 X] and PIK3R1 [p. R574del] in the second group, and KRAS [p. G12V] in G13) (Fig. 7f).

We investigated the clonal populations in the 13 glands by using PyClone[46]. The somatic SNVs were classified into 11 clusters (Supplementary Fig. 10a). Two clusters consisted of public mutations that were shared among all the glands in either of the two groups. Six clusters encompassed partially shared mutations among some glands in a group. Three clusters contained private mutations unique to single glands. Then, we performed clone ordering of the identified clusters with ClonEvol[47]. The fish plots showed that ancestral clones (A and G) characterized by public mutations initially expanded, and then their descendant clones characterized by partially shared or private mutations emerged (Fig. 7g). Some of the partially shared mutations that reached fixation in vertical glands were observed in subclonal states in their respective rhizomes (Fig. 7g, h). The most recent clonal expansions in the first and second groups were estimated to have occurred at ages 34.5 (95% CI, 33.0–36.0) and 35.1 (95% CI, 33.7–36.6), respectively (Fig. 7i). These results imply that the two rhizome structures were independently shaped by clonal expansions of ancestral clones approximately 9 years ago, and stem/progenitor cells with self-renewal and differentiation capabilities residing in different zones of the rhizomes

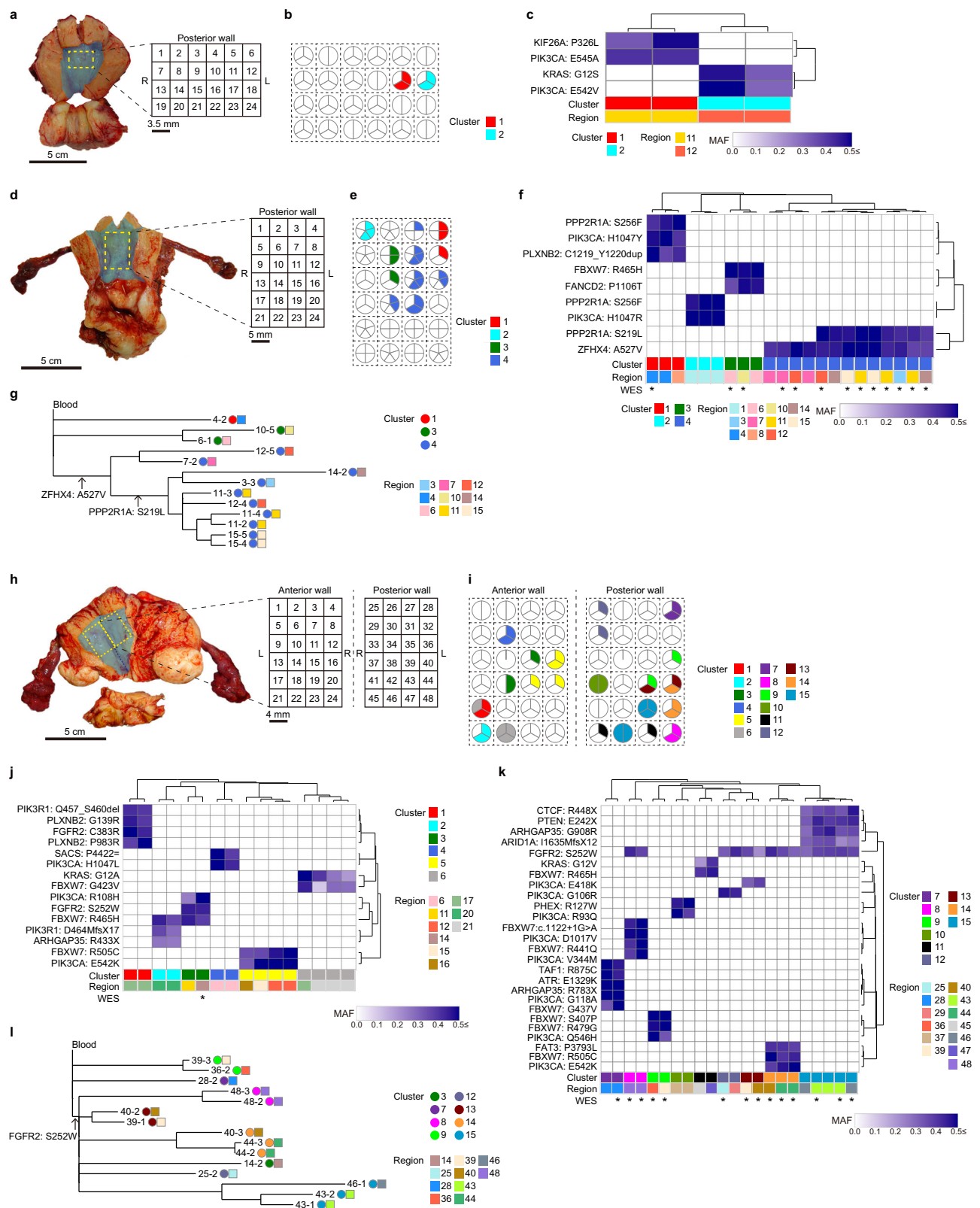

acquired their own mutations and subsequently gave rise to vertical glands.

Next, we focused on the clonal evolution of three glands (G3–G5) that diverged near the center of the functional layer (Fig. 7c). From the background of an ancestral clone (A), two distinct mutant clones (C and D) proliferated to generate the lower glands (G3 and G4, respectively). Both of these mutant clones coexisted in subclonal states in the upper gland (G5) (Fig. 7g, h). Specifically, the MAFs of mutations observed in either G3 or G4 were close to 0.5 in the corresponding glands but approximately 0.25 in G5 (Supplementary Fig. 10b). These results suggest that G3 and G4 independently rose from the rhizome (G1), and G5 was formed by the confluence of G3 and G4 on their way to the luminal side.

**Fig. 4 Sequencing of spatially resolved single endometrial glands.** The results for three subjects are represented in different panels: **a–c** 38-year-old woman with endometriosis (subject 33), **d–g** 41-year-old woman with myoma uteri (subject 34) and **h, i** 46-year-old woman with cervical carcinoma in situ (subject 35). The anterior and posterior walls of the endometrium from subject 35 were used. **a, d, h** Macroscopic images of uterine tissue samples obtained from three subjects who underwent a total hysterectomy, in which the endometrium is highlighted in light blue. Schematic layouts show the partitioning of the resected endometrium into square grids with their identification numbers. R: right side of uterus. L: left side of uterus. **b, e, i** Clusters detected by hierarchical clustering analyses mapped by projecting on the schematic layout of the endometrium sample. Pie charts show numbers and proportions of glands grouped in the clusters. The colors of clusters are the same as in (**c, f, j, k**). **c, f, j, k** Hierarchical clustering of the mutation profiles of endometrial glands based on MAFs for a set of high-MAF mutations shared among at least two glands in the target-gene sequencing. Glands marked with an asterisk were used for WES. Color density indicates the MAF of each somatic mutation. **g, l** Phylogenetic tree for binary mutation profiles based on WES.

## Discussion

Here, we showed that the genomes of normal endometrial glands were suffused with mutations caused by clock-like mutational processes and oxidative stress. The burdens of substitutions relevant to these mutation signatures were observed to increase with age and CNMCs, which is consistent with recent studies[16,17]. The highly regenerative nature of the endometrium may be involved in the accumulation of somatic mutations. Most of the genes showing evidence of positive selection in the normal endometrium are frequently mutated in endometrial cancer[39,48], suggesting that mutations in the genomes of endometrial cancer are inherited from the normal endometrium. Although cancer-associated gene mutations are pervasive in endometrial glands, ~3% of women are estimated to be diagnosed with uterine cancer during their lifetime, suggesting that cancer-associated gene mutations alone are not sufficient to initiate cancer. DNA quality control pathways are thought to be permissive of mutagenesis in normal cells because the repair of DNA lesions requires time and cellular resources; therefore, cells maintaining comprehensive repair would sacrifice their cellular survival[49]. Moreover, there is a possibility that some cancer-associated gene mutations might enhance cellular proliferation and benefit endometrial regeneration after menstruation, as shown in liver regeneration[50].

We demonstrated that glands originating from the same mutant clones were distributed across spatially separated regions of the endometrium. The extent of the colonization by mutant clones differed across endometrium samples. In the endometrium from a 50-year-old woman, two distinct mutant clones occupied substantial areas, one of which had a frameshift insertion in *PTEN* accompanied by arm-level CN-LOHs and had undergone rapid clonal expansion. Seemingly, this clone might have been only a few steps away from turning malignant. However, there was a long latency between the CN-LOHs and the most recent clonal expansion, implying that even arm-level copy number alterations are not sufficient to immediately trigger tumorigenesis. Our finding corroborated recent studies in kidney and lung cancers showing structural rearrangements that occurred during childhood and adolescence, many decades before disease diagnosis[43,44]. Nongenetic factors such as epigenetic alterations and complex interplays with stromal cells, hormonal influence and the immune system may also contribute to the transformation of normal cells into malignant cells[8].

The missing piece is how incipient mutant clones colonize normal tissues[26]. Clonal expansions in normal tissues may be constrained by anatomical features of the corresponding tissues; therefore, tissue-specific mechanisms should be elucidated[51]. Clonal expansion by crypt fission is a striking anomaly in the unraveled tissue-specific mechanism, where mutations that occur in a colonic stem cell are fixed in a whole crypt by monoclonal conversion and then the wholly mutated crypt divides into two mutated daughter crypts[52,53]. To approach the endometrium-specific mechanisms of clonal expansion, we leveraged 3D imaging analysis to inspect the continuum of glandular epithelium in the normal human endometrium. Although the presence of basal glands running horizontally and parallel to the luminal surface has been documented since at least the 1920s[54,55], plexus rhizome-like structures, in which multiple vertical glands were linked with the horizontal basal gland, were discovered only recently with the advent of 3D imaging analyses[30,31]. Tempest et al. have proposed that rhizome-structures assist self-preservation, self-renewal, and scarless regeneration of the human endometrium as a niche of endometrial epithelial stem/progenitor cells[30]. In this study, we demonstrated that the continuum of a rhizome and vertical glands had a monoclonal origin. Taking into consideration the widely accepted idea that the functional layer regenerates from the glands remaining in the basalis after menstruation[56,57], we propose a plausible model of clonal expansion in the normal endometrium. Residual basal glands extend horizontally along the muscular layer to shape monoclonal rhizomes. Then, the rhizome gives rise to vertical glands that have the same clonal origin. Some rhizome structures persist for many cycles of repair and regeneration during menstrual cycles and further expand their territories. Several rounds of clonal conversion might occur when new clones acquire selective advantages by cancer-associated gene mutations. At the same time, stem/progenitor cells resident in the rhizome acquire unique mutations, which leads to local variability of the vertical glands. The clonal expansion through rhizome structures proposed in this study is by no means the only mechanism. Since rhizome structures are thought to be specific to animals that menstruate[30–33], the development of rhizome structures and clonal expansion through rhizome structures might be byproducts of the evolution of menstruation. The elucidation of the mechanisms of clonal expansion in both menstruating and nonmenstruating species will provide fundamental clues for understanding homeostatic tissue remodeling and oncogenesis. Another important finding is the admixed genomic composition of branched glands, which implies that two glands from a rhizome join together and open into the superficial layer in a bottom-up manner. A previous study reported individual endometrial glands containing both cytochrome c oxidase (CCO)-positive and CCO-negative cells[30]. Our findings give an important perspective on the monoclonal and polyclonal cellular compositions of single endometrial glands.

The mechanisms of rhizome structure formation in the human endometrium are totally unknown. Anatomical, embryological, and physiological studies are required to elucidate when and how the rhizome structures develop. For this purpose, it will be helpful to clarify the spatial relationship between rhizome structure and vascular network by 3D imaging. The biological and medical significance of the rhizome structures also remains obscure. We speculate that the rhizome structures act as a double-edged sword. The presence of the rhizome structures might be beneficial for post-menstrual endometrial repair by protecting the endometrial stem/progenitor cells from shedding at the menstrual phase[30,31]. We presume that clones with cancer-associated gene

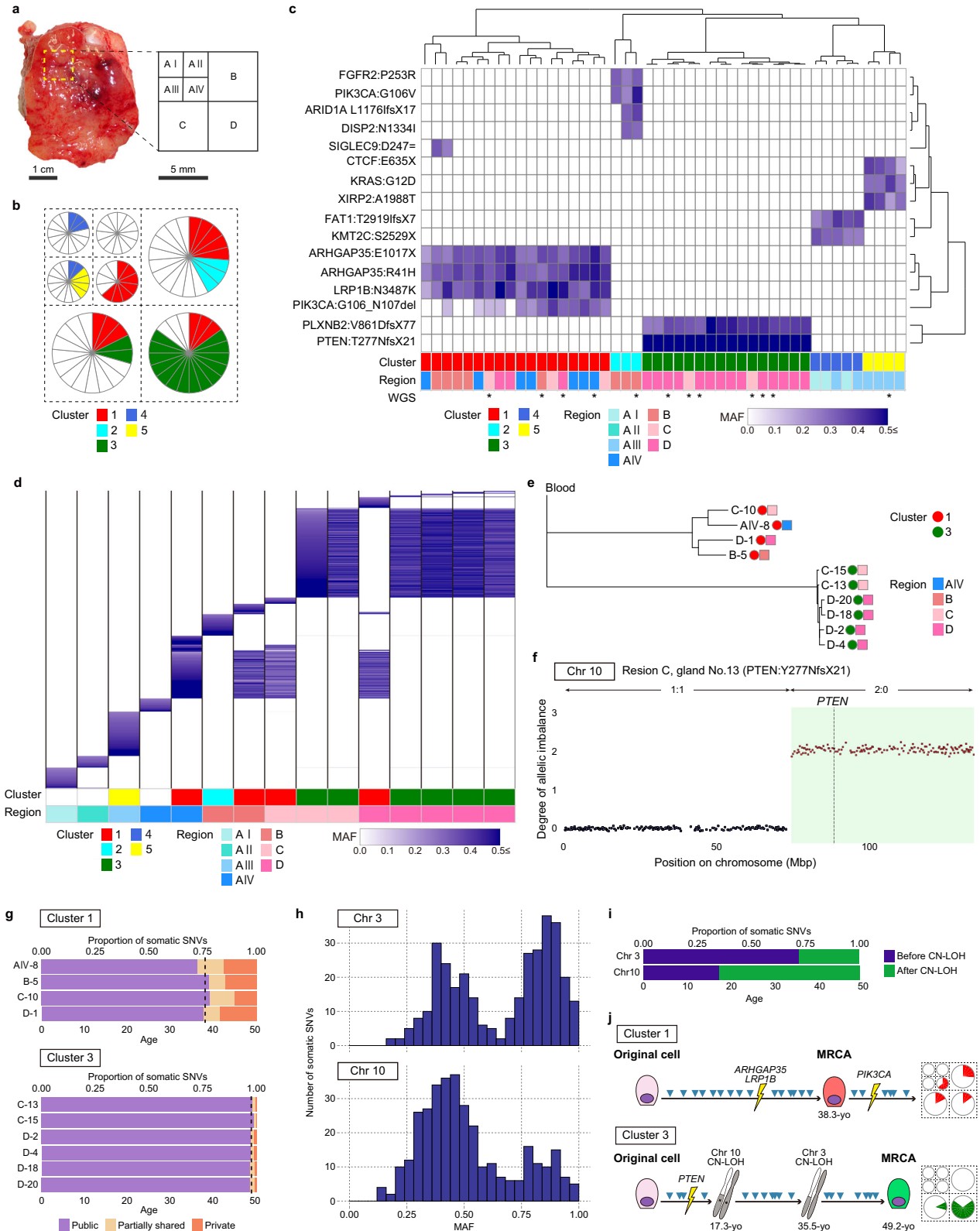

mutations may confer a proliferative advantage and contribute to stable tissue regeneration by expanding the area of rhizome structure. On the other hand, the rhizome structures might confer deleterious effects to predispose women to endometrium-related diseases by accumulating cancer-associated gene mutations. A recent study showed that adenomyosis and histologically normal endometrium adjacent to the adenomyotic lesions had identical

*KRAS* hotspot mutations, suggesting that *KRAS*-mutated adenomyotic clones originate from normal endometrium[58]. In the WES study for a patient with ovarian clear cell carcinoma with concurrent endometriosis, we showed that many somatic mutations including cancer-associated gene mutations were shared among epithelium samples from uterine endometrium, endometriotic lesions distant from and adjacent to the carcinoma, and the

**Fig. 5 Spatiotemporal evolution of mutant clones in the normal endometrium. a** A macroscopic image of a part of a tissue sample of the uterus obtained from a 50-year-old woman (subject 36) with myoma uteri who underwent a total laparoscopic hysterectomy. A part of the endometrium enclosed by the dashed yellow line was intensively analyzed. A schematic layout represents partitioning of the resected endometrium into square grids with their identifiers. **b** Clusters detected by hierarchical clustering analysis are mapped by projection onto the schematic layout of the endometrium sample. Pie charts show the numbers and proportions of glands grouped in the clusters. The colors of the clusters are the same as in (**c**). **c** Hierarchical clustering of mutation profiles of a set of high-MAF mutations shared among at least two glands in the target-gene sequencing. Glands marked with an asterisk were used for WGS. Color density indicates the MAF of each somatic mutation. **d** Sharing pattern of somatic SNVs based on WGS among 15 glands. Color density indicates the MAF of each somatic mutation. **e** Phylogenetic tree for binary mutation profiles based on WGS. **f** CN-LOH at chromosome 10 encompassing *PTEN*. The profile for a representative gland (No. 13 in region C) is shown. The region affected by CN-LOH is highlighted in light green. The numbers separated by a colon are the major and minor copy numbers for the regions indicated by the arrows. **g** Chronological ages at which clonal expansions occurred. The dashed line corresponds to the average of the proportion of public mutations to the overall mutational burden. **h** Bimodal distributions of the MAFs of public mutations in the regions affected by two CN-LOHs. **i** Chronological ages at which two CN-LOHs occurred in the mutant clones of cluster 3. **j** Schematic illustration of genomic evolution in the normal endometrium. MRCA, most recent common ancestor. Source data are provided as a Source data file.

carcinoma[59]. As clonal hematopoiesis increases the risks for blood cancer and cardiovascular disease[10], there is a possibility that the overrepresentation of endometrial glands derived from a single clone raise the risks for endometrium-related diseases. Long-term follow-up studies are needed to determine whether mutant clones with cancer-associated gene mutations extends their territories over time and whether the situation where the population of endometrial epithelial cells is dominated by a small number of mutant clones increases the risks for endometrium-related diseases such as adenomyosis, endometriosis and endometrial cancer. Several lines of evidence shows that oral contraceptives decrease the risks for endometrial and ovarian cancers[60] and endometriosis[61]. The biological mechanism behind these preventive effects may be partially explained if we demonstrate that the regulation of the menstrual cycle by oral contraceptives alleviates the mutational burden and the aberrant proliferation of the rhizome structure in the endometrium. Further efforts are needed to unlock the mystery of the rhizome structures.

We demonstrated the somatic evolutionary dynamics of mutant clones in the human endometrium at a microscopic spatial resolution. Three-dimensional mapping of mutant clones in the endometrium will illuminate the path toward a more precise understanding of the mechanisms of endometrial regeneration during the menstrual cycle and the development of therapies for the prevention and treatment of endometrium-related diseases.

## Methods

**Human sample collection**. This study was approved by the institutional ethics review boards of Niigata University (G2017-0010 and G2019-0038), Nagaoka Chuo General Hospital (331), National Institute of Genetics (29-15), and Sasaki Institute (ER2020-04). We recruited study participants at the Niigata University Medical and Dental Hospital and the Nagaoka Chuo General Hospital between December 2015 and September 2019. All subjects provided written informed consent for the collection of samples and analyses, and for the publication of their clinical information.

We collected 1087 single uterine endometrial gland samples from 32 patients (aged 21–53 years) with no endometrial gynecological disease (Subject No. 1–32). Among them, 196 glands were excluded at the quality control step of sequence data analysis. The samples were from endometrial biopsies using suction catheters in gynecological patients ($n = 23$) or endometrial curettage in patients undergoing hysterectomy ($n = 9$). Endometrial suction was performed under lumbar anesthesia or general anesthesia during surgery for gynecological disease. Endometrial curettage was performed on surgically resected uteri using a disposable scalpel. Diagnosis of the sample cases included myoma uteri ($n = 11$), ovarian dermoid cyst ($n = 7$), adenomyosis or ovarian endometriosis ($n = 9$), cervical intraepithelial neoplasia (CIN)3 or carcinoma in situ (CIS) ($n = 5$), cervical cancer ($n = 1$) and ovarian clear cell carcinoma ($n = 1$) (there was some overlap). Clinicopathological information is shown in Supplementary Table 1.

We performed multisegmental sampling for a total of four patients (aged 38–50 years) who underwent hysterectomy (Subject No. 33–36). Diagnosis of the sample cases included myoma uteri ($n = 2$), ovarian endometriosis ($n = 1$), and CIN3 or CIS ($n = 1$). We divided the endometrium into 7–48 segments and selected 451 single endometrial gland samples, with a sampling depth range of

3–20 samples per segment. The area of each segment was 6.25–25 mm². Among them, 44 glands were excluded at the quality control step of sequence data analysis. Clinicopathological information is shown in Supplementary Table 2. For the 3D imaging analysis, we collected four uterine endometrial samples from 30 to 52-year-old premenopausal women who underwent hysterectomy (Subject No. 37–40). Diagnosis of the samples included myoma uteri ($n = 2$) or cervical cancer IA1 ($n = 1$) or IB1 ($n = 1$). Clinicopathological information is shown in Supplementary Table 3. For the WGS analysis using laser microdissected tissues, we collected uterine endometrial samples from a 43-year-old premenopausal woman who underwent hysterectomy because of endometriosis (Subject No. 41).

We confirmed that the endometrial tissues collected from all the individuals were clinically and histologically normal. Peripheral blood samples were also collected from each patient and used as matched controls for the delineation of somatic versus germline variations.

**Isolation of single endometrial glands**. The isolation of single endometrial glands from bulk endometrial tissue was performed as described in our previous studies[15,29], with some modifications. Briefly, minced fresh endometrial tissues, consisting of endometrial glands and stroma, were placed in Dulbecco's modified Eagle's medium (Thermo Fisher Scientific) containing 180 U/ml collagenase type 3 (Worthington Biochemical Corporation). After shaking gently at 37 °C for 40 min, this solution of collagenase-digested tissue was poured into a 40-micron EASY-strainer (Greiner Bio-One), and cold phosphate-buffered saline (PBS) was poured over the screen to wash away the digestion-loosened endometrial stromal cells. The tissue remaining on the strainer, mostly epithelial cell glands, was transferred into a cell culture dish. After gentle pipetting, the separated endometrial glands sank to the bottom of the dish. Individual glands were picked up precisely under a microscope. Each gland was collected into a 0.2 ml microtube containing ATL buffer (QIAGEN) and stored at −80 °C until DNA extraction. A total of 1538 glands from 36 patients were used for target-gene sequencing analyses. Thirteen glands from subject No. 34 and 15 glands from subject No. 35 were also used for WES analysis. Thirteen glands from subject No. 36 were also used for WGS analysis.

**Laser-microdissection of each entire gland**. All endometrial tissues were immediately separated from the surgical specimen, embedded in Tissue-Tek O.C.T. compound (Sakura Finetek) in a Tissue-Tek Cryomold (Sakura Finetek), frozen in liquid nitrogen, and stored at −80 °C. We cut 12-μm-thick serial frozen sections with a Cryotome FSE (Thermo Fisher Scientific) and mounted them on PEN-Membrane Slides (Leica).

For laser microdissection, the cryosections were fixed with 100% methanol for 3 min and then stained with toluidine blue for 30 s. Before laser microdissection, all images of cryosections were stored using the Specimen Overview function of the LMD7 laser microdissection microscope (Leica) to assess glandular continuity. We performed laser microdissection using LMD7 (Leica), distinguishing the vertical and horizontal glands, which often branched from or connected with each other (Fig. 7). The isolated epithelial tissues of each gland were collected in the caps of 0.2 ml PCR tubes (Axygen). The median number of frozen sections for sampling each gland was 13 sections (range: 8-18).

**DNA extraction**. DNA extraction from the isolated endometrial glands and the laser-microdissected glands was performed using a QIAamp DNA Micro Kit (QIAGEN) according to the manufacturer's protocol. Case-matched control DNA was extracted from peripheral blood with a QIAamp DNA Blood Maxi Kit (QIAGEN) according to the manufacturer's instructions. All purified DNA samples were stored at −80 °C until subsequent analyses.

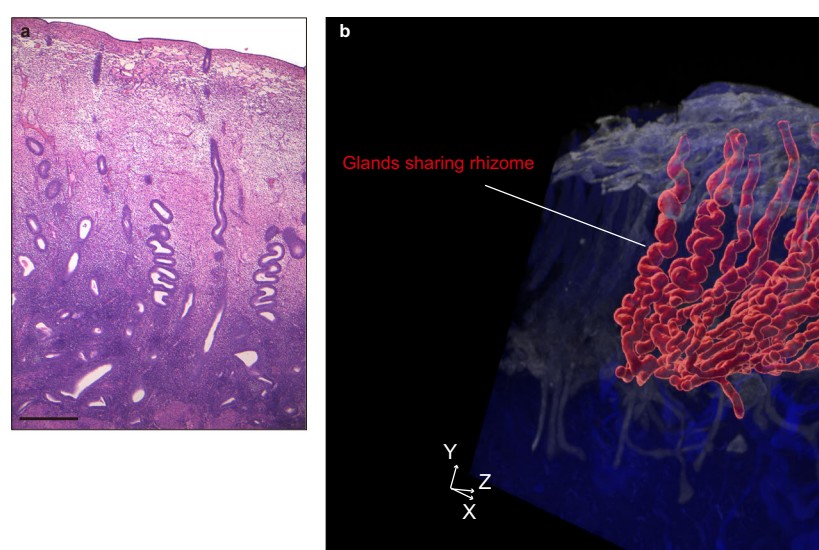

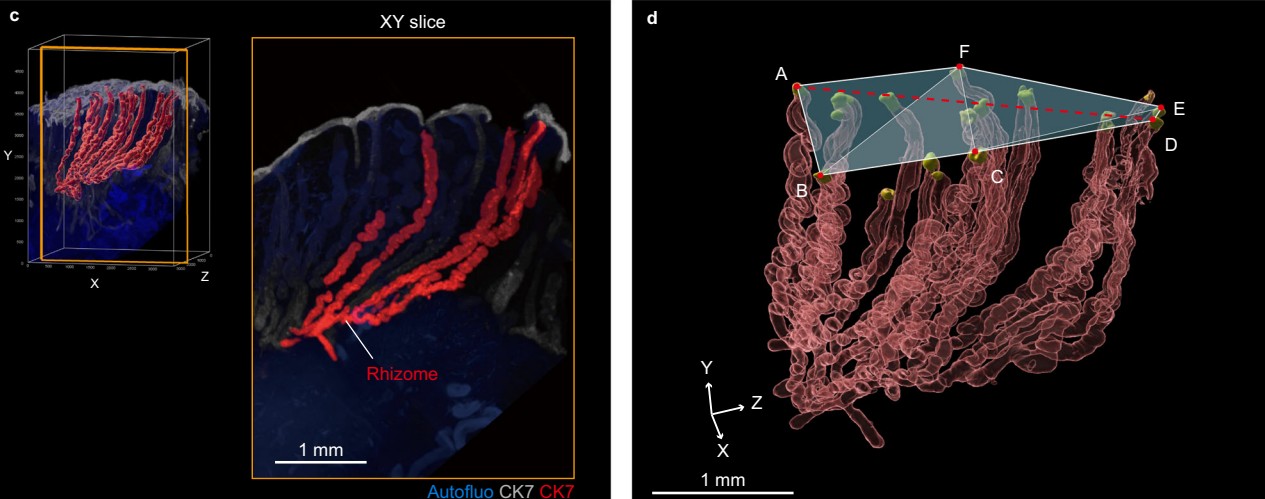

**Fig. 6 3D imaging of normal proliferative endometrial tissue. a** Microscopic hematoxylin- and eosin-stained images of endometrium obtained from a 52-year-old woman (subject 40) with myoma uteri who underwent a total hysterectomy. Scale bar, 500 μm. **b** Reconstructed 3D image of the endometrial tissue. The red object represents the 3D structure of the glands sharing a rhizome. **c** 3D and XY cross-sectional images capturing a rhizome structure that runs horizontally along the muscular layer and gives rise to multiple vertical glands. From the reconstructed 3D image (left panel), XY cross-sectional images (right panel) with a slice thickness of 200 μm are extracted. The XY slice is enclosed by orange. **d** 3D image quantifying the longest distance between glands sharing the rhizome (red line) and the area occupied by the glands (light blue area). Red object: 3D image of the structures of the glands sharing a rhizome. Yellow objects: the tips of the glands. 3D images were obtained by light-sheet fluorescence microscopy. Autofluorescence and CK7-expressing endometrial epithelial cells were measured by excitation with 488 nm and 532 nm lasers, respectively. Red and yellow objects were made by the Surface module in Imaris software. Autofluo, autofluorescence; CK7, cytokeratin 7. Source data are provided as a Source data file.

**Target-gene sequencing.** The target-gene sequencing of single endometrial glands for 112 genes was performed as described in our previous studies with some modifications[15,59,62,63]. Briefly, 112 genes were selected (Supplementary Data) based on WES data for ovarian endometriosis and normal uterine endometrium[15], the mutation profiles in endometriosis-related ovarian cancer[64] and in endometrial cancer[48], and genes involved in DNA repair pathways[65,66].

DNA samples were fragmented using a KAPA Frag Kit (KAPA Biosystems). Sequencing libraries were constructed with a NEBNext Ultra II DNA Library Prep Kit for Illumina (New England Biolabs). Libraries of up to 96 samples were pooled in equimolar amounts and then hybridized to probes of a SeqCap EZ Prime Choice System (Roche Diagnostics) in a single enrichment reaction. The DNA probe set was selected by using NimbleDesign (Version 3.8) (http://design.nimblegen.com). The quantity and size distribution of the captured libraries were assessed by a Qubit 2.0 Fluorometer (Thermo Fisher Scientific) and Bioanalyzer 2100 (Agilent Technologies), respectively. The libraries were then sequenced via the Illumina HiSeq 2500, HiSeq 4000 or NovaSeq 6000 platform with 2×100-bp or 2×150-bp paired-end modules (Illumina).

**Data preprocessing.** As a quality control step, the Illumina adapter sequences were trimmed by using Trim Galore (Version 0.6.3) (https://www.bioinformatics.babraham.ac.uk/projects/trim_galore/). Low-quality sequences were excluded or trimmed with Trimmomatic (Version 0.39)[67]. The filtered sequence reads were aligned to the human reference genome (GRCh38) containing sequence decoys and virus sequences generated by the Genomic Data Commons (GDC) of the National Cancer Institute (NCI) using BWA-MEM (Version 0.7.17)[68,69]. The sequence alignment map (SAM) files were sorted and converted to the binary alignment map (BAM) file format with SAMtools (Version 1.9)[70]. The BAM files were processed using Picard tools (Version 2.20.6) (http://broadinstitute.github.io/picard/) to remove PCR duplicates. Base quality recalibration was conducted using GATK (Version 4.1.3.0)[71,72]. The average depths and the coverages of the target regions were calculated with SAMtools. BEDOPS (Version 2.4.36)[73] and BEDTools (v2.28.0)[74] were used in the handling of FASTA, VCF and BED files.

We used endometrial glands for subsequent analyses if more than 70% of the target bases were covered by at least 20 reads. For 32 subjects used in the analyses

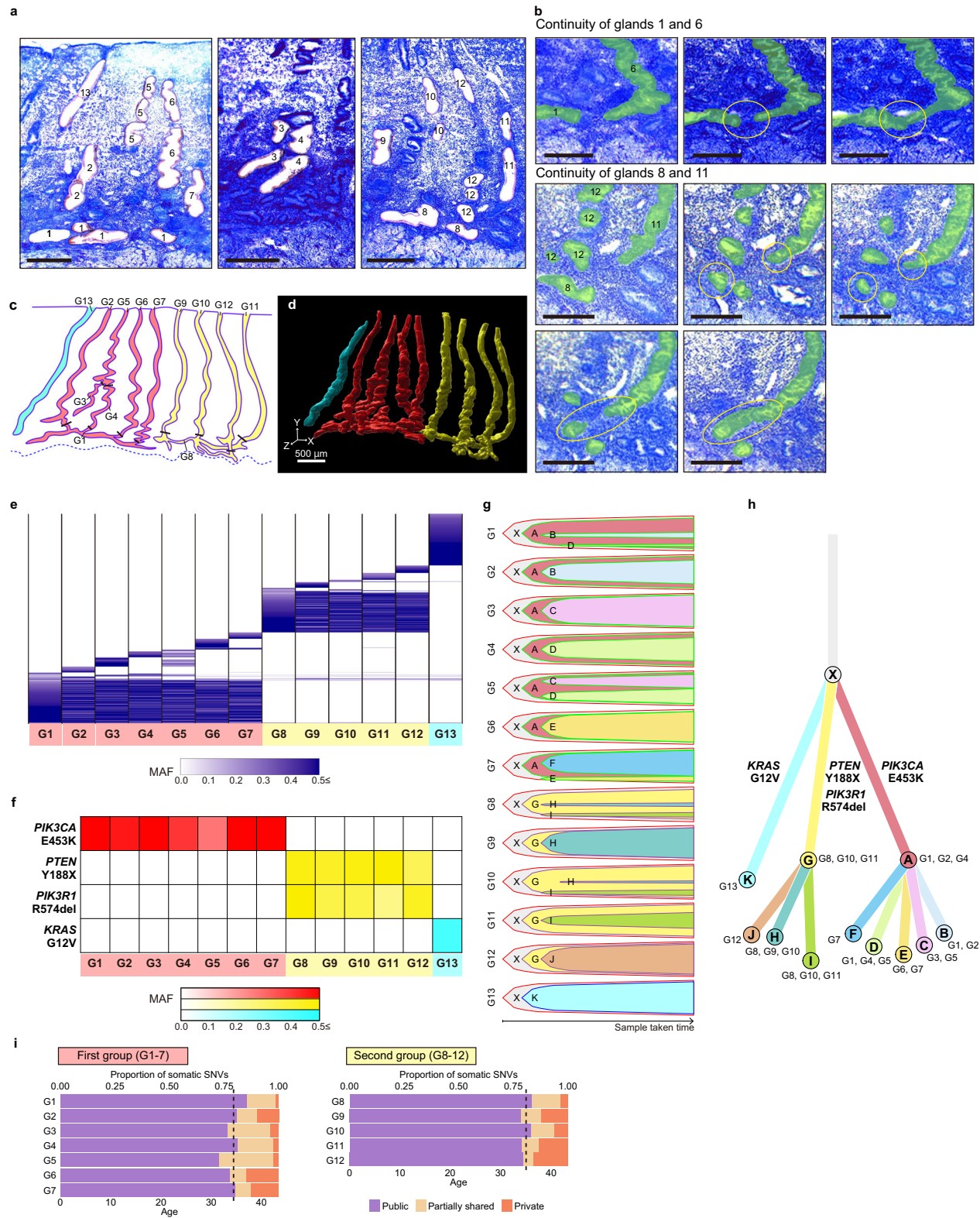

of mutational burden, mutational signatures and selections acting on cancer-associated gene mutations, the means of the average sequencing depth and the coverage at ≥20 reads for the target regions were 87.7 and 93.5%, respectively. For four subjects used in the analysis of spatially resolved single endometrial gland sequencing, the means of the average sequencing depth and the coverage at ≥20 reads for the target regions were 80.9 and 93.1%, respectively.

**Variant calling.** Somatic SNVs and short indels were called in each pair of endometrial gland and matched blood samples by using Strelka2 (Version

2.9.10)[75]. For somatic indel calling, we utilized the information about candidate indel sites provided by Manta (Version 1.6.0)[76]. The variants whose empirical variant scores provided by Strelka2 were greater than 13.0103 (= −10 × log₁₀ 0.05) were used for subsequent analyses. In addition, we excluded variants whose frequencies were greater than or equal to 0.001 in any of the general populations from the 1000 Genomes Project[77], the National Heart, Lung, and Blood Institute (NHLBI) GO Exome Sequencing Project[78], and the Genome Aggregation Database (gnomAD)[79] to prevent false-positive variant calls. Functional annotations for the protein-coding and transcription-related effects of the identified variants were implemented by Ensembl Variant Effect Predictor (VEP)[80]. Curated information

**Fig. 7 Genomic evolution of endometrial glands connected through rhizome structures. a** Toluidine blue-stained images of endometrial glands from a 43-year-old woman after laser microdissection. Scale bar, 500 μm. **b** Evaluation of the continuity between glands by serial section images before laser microdissection. Top images, continuity between glands 1 and 6. Middle and bottom images, continuity between glands 8 and 11. Scale bar, 300 μm. **c** Schematic illustration of glands isolated by laser microdissection. According to their continuity, the first group (rhizome: G1, and vertical glands: G2 to G7), the second group (rhizome: G8, and vertical glands: G9 to G12), and an independent vertical gland (G13) are color-coded in red, yellow, and light blue, respectively. **d** Reconstructed 3D image of glands isolated by laser microdissection, based on serial section images. The color code assignment is as described in (**c**). **e** Sharing pattern of somatic SNVs based on WGS among 13 glands. Color density indicates the MAF of each somatic mutation. **f** Heatmap of the prevalence of representative cancer-associated gene mutations in 13 glands. Color density indicates the MAF of each somatic mutation. **g** Fish plots showing the clonal evolution of 11 mutant clones (A-K) within 13 glands. **h** Branch-based consensus clonal evolution tree of 11 mutant clones (A-K) among 13 glands. Nodes and edges correspond to mutant clones and somatic mutations accumulated during evolution between the connected clones, respectively. The identifiers of glands are indicated beside a node if the clone was observed in the corresponding glands at the time when the sample was taken. **i** Chronological ages at which clonal expansions occurred in mutant clones of the first and second groups of glands. The dashed line corresponds to the average proportion of public mutations to the overall mutational burden in each cluster. Source data are provided as a Source data file.

---

about cancer-associated genes and their functional roles in cancer development was retrieved from the COSMIC database[38].

**Mutational signature analysis.** We used the identified somatic SNVs with high MAF (≥0.25) from the 32 subjects for mutational signature analysis. The somatic SNVs with high MAF were classified into 96 mutation classes defined by the six pyrimidine substitutions (C>A, C>G, C>T, T>A, T>C, and T>G) in combination with the flanking 5′ and 3′ bases. In our mutational signature analysis, we fitted the 96-mutation catalog to a predefined list of known signatures[81,82]. We did not select an approach for de novo signature extraction because the number of somatic mutations was not large enough in this study. As a reference set of known mutational signatures, we used the COSMIC mutational signatures version 3[34]. We selected a total of eleven SBS signatures (SBS1, SBS2, SBS3, SBS5, SBS10a, SBS10b, SBS13, SBS15, SBS18, SBS40, and SBS44) whose activities were estimated to be present in at least 10% of samples in any of three gynecologic cancers (cervical cancer, endometrial cancer and ovarian cancer) based on the COSMIC mutational signatures (https://cancer.sanger.ac.uk/cosmic/signatures/). We implemented a fitting approach by using sigfit[83]. We ran four Markov chains with a total of 50,000 iterations, including a burn-in of 25,000 samples. We estimated the highest posterior density (HPD) interval for each of the SBS signatures. If the 90% lower end of the HPD interval for a SBS signature was above the threshold (0.01, default value), we considered that the SBS signature was a significantly active signature.

**Calculation of mutational burden.** Mutational burden ($b_i$) was calculated for the $i$ th subject over endometrial glands as follows:

$$b_i = \sum_j n_{ij} \times 10^6 / \sum_j l_{ij}, \quad (1)$$

where $n_{ij}$ and $l_{ij}$ are the number of somatic mutations and the number of bases within the target region that were covered by at least 20 reads in the $j$ th gland of $i$ th subject, respectively. We assessed the association between the burden of somatic mutations and the clinical features of the subjects (Supplementary Table 1). The Pearson correlation coefficient and one-way analysis of variance were used for quantitative and categorical clinical variables, respectively. Linear regression analysis was conducted to assess the effects of clinical features on the burden of somatic mutations with adjustment for age. The statistical tests were performed as two-tailed tests using the stats package of the R software (https://www.R-project.org/).

**Calculation of the burden of driver mutations.** We selected cancer driver genes that were included in the Cancer Gene Census[40] and the pan-gynecologic cancer-associated genes[41] from the 112 genes (Supplementary Table). As a result, the nine genes were selected: *ARID1A*, *CTNNB1*, *FBXW7*, *KRAS*, *PIK3CA*, *PIK3R1*, *PPP2R1A*, *PTEN*, and *TP53*. The burden of driver mutations is defined as the number of non-silent mutations in these nine genes per gland in each subject. The difference in the average age or CNMCs between carriers of *KRAS* mutations exhibiting allelic imbalance (MAF ≥ 0.8) and non-carriers was examined by Wilcoxon rank-sum test with the exactRankTests package of the R software.

**Identification of genes under positive and negative selection.** We searched for genes under the pressures of positive and negative selection based on the dN/dS ratio. To estimate the dN/dS ratio, we patterned our approach after the Poisson framework developed by previous studies[39,84]. We modeled the numbers of SNVs resulting in missense ($n_i^m$), nonsense ($n_i^n$) and synonymous ($n_i^s$) substitutions in the $i$ th trinucleotide context as follows:

$$n_i^m \sim \text{Poisson}(\lambda = t \times r_i \times L_i^m \times \omega^m), \quad (2)$$

$$n_i^n \sim \text{Poisson}(\lambda = t \times r_i \times L_i^n \times \omega^n), \quad (3)$$

$$n_i^s \sim \text{Poisson}(\lambda = t \times r_i \times L_i^s), \quad (4)$$

where $t$ is the baseline substitution rate per site, $r_i$ is the relative substitution rate in the $i$ th trinucleotide context, $L_i$ is the number of sites at which the $i$ th substitution results in (m)issense, (n)onsense, and (s)ynonymous mutation, and $\omega$ is the dN/dS ratio. The number of identified somatic mutations in this study was not large enough to estimate substitution rates in trinucleotide contexts. Therefore, we relied on the COSMIC database as an unbiased catalog of somatic mutations. We downloaded the file "CosmicNCV.tsv" of v94 (May 2021). First, we retrieved non-coding somatic SNVs from the file by using the following filtering criteria: (i) inclusion of SNVs confirmed to be somatic, and (ii) exclusion of known single nucleotide polymorphisms. Then, we selected the SNVs if the primary site of the sample with the SNV was cervix, endometrium or ovary. As the consequence, about 1.1 million somatic SNVs in noncoding regions from the COSMIC database were classified into 96 mutation classes constituted by the six pyrimidine substitutions in combination with the flanking 5′ and 3′ bases. Then, the relative substitution rate for each trinucleotide context ($r_i$) was calculated. The baseline substitution rate per site ($t$) was selected to satisfy the following equation:

$$\sum_i \sum_{j \in \{m,n,s\}} n_i^j = \sum_i \sum_{j \in \{m,n,s\}} t \times r_i \times L_i^j. \quad (5)$$

The values $n^m$, $n^n$ and $n^s$ were evaluated for several sets of genes or for each gene. We considered three sets of genes: (i) all 112 genes, (ii) 48 genes included in the Cancer Gene Census[40], and (iii) 15 genes included in the pan-gynecologic cancer-associated genes[41]. Poisson regression analysis was implemented, where $n^m$ (or $n^n$) and $n^s$ were modeled by including the log-transformed value of their expected numbers (i.e., $\sum_i t \times r_i \times L_i$) as the offset and the $\omega$ term. When examining the sets of genes, the significance of $\omega$ was assessed by Poisson regression analysis. According to a previous study[39], the fraction of genuine driver mutations in a group of genes was calculated as follows: $f = (\omega - 1)/\omega$.

When the signatures of positive selection were evaluated at the level of individual genes, we identified 27 genes with at least five SNVs with high MAF in their coding sequences. Additionally, we considered genes if the observed numbers of missense ($n^m$ or $n^n$) were larger than their expected numbers calculated by using $t$, $r_i$ and $L_i$. Then, the likelihood ratio test was conducted to compare the two Poisson regression models with and without the $\omega$ term. To account for multiple hypothesis testing, genes satisfying a false discovery rate based on Benjamini–Hochberg procedure < 0.1 were considered to be significant[85]. The Poisson regression analyses were performed by the glm function from the MASS package in the R environment.

**Analyses of spatially resolved single endometrial gland sequencing.** To evaluate the extent to which somatic mutations were shared between glands located in spatially separated regions of the endometrium, we compiled MAF profiles of all the mutation sites for all the glands by counting the sequence reads supporting the reference and mutant alleles with SAMtools mpileup[70]. For this analysis, the reads mapped with high confidence (mapping quality > 30) were used. Then, the allele-specific counts were measured by using only high-confidence base calls (base quality > 20) at the mutation sites. We excluded sites whose MAFs in the matched blood sample exceeded 0.05. We selected informative mutations based on the following criteria: (i) a set of mutations that were shared among a group of glands at MAF of greater than or equal to 0.10; and (ii) mutations with MAF values greater than or equal to 0.25 in at least two glands. The MAF profiles of the informative mutation sites were analyzed by hierarchical clustering analysis with the superheat R package[86].

**Experimental procedures for WES and WGS.** The libraries were prepared as described above. For WES, we used a hybridization capture method with IDT xGen Exome Research Panel v2 (IDT xGen Exome Research Panel v2 (Integrated DNA Technologies), in which precapture libraries from at most six samples were pooled

and then hybridized in a single enrichment reaction. For the WGS of endometrial glands isolated by laser microdissection, DNA samples were repaired by using NEBNext FFPE DNA Repair Mix (New England Biolabs). The precapture libraries were prepared as described above and used for a subsequent sequencing step. The libraries were sequenced via the Illumina NovaSeq 6000 platform with a $2 \times 150$-bp paired-end module (Illumina).

**Bioinformatics pipeline for WES and WGS.** In WES, we used target regions in the BED format provided by the manufacturer. The means of the average sequencing depth and the coverage at ≥10 reads for the target bases over 30 WES data were 49.0 and 96.6%, respectively.

To filter somatic mutations in WGS, we used a "universal mask" outlined in previous studies[87–89]. The universal mask encompasses regions in which false positive variants are recurrently detected. Low-complexity regions in the human reference genome (GRCh38) were determined by mdust (https://github.com/lh3/mdust). DNA satellites and low complexity regions were based on the RepeatMasker track from the UCSC Genome Browser (https://genome.ucsc.edu/). Homopolymeric stretches (≥7 bp) were sought by seqtk (https://github.com/lh3/seqtk). The low-mappability mask was generated by following the SNPable Regions procedure (http://lh3lh3.users.sourceforge.net/snpable.shtml). Briefly, at each position in the human reference genome (GRCh38), all possible 75-mers overlapping the position were extracted and mapped back to the reference genome with BWA. Low mappability regions were defined as genomic positions for which 37 or fewer overlapping 75-mers were mapped elsewhere with at most one mismatch or gap. The means of the average sequencing depth and the coverage at ≥10 reads for the target bases over 30 WGS data were 27.2 and 96.1%, respectively.

To identify somatic variants with high confidence, we used the following criteria: (i) the empirical variant score provided by Strelka2 was greater than 13.0103; (ii) the frequencies in the abovementioned general populations were smaller than 0.001; (iii) the sequencing depth was greater than or equal to 20; (iv) eight or more reads supported the mutant allele; (v) the MAF was greater than or equal to 0.25; and (vi) the MAF in the matched blood sample did not exceed 0.05.

**Detection of somatic copy number alterations.** Somatic copy number alterations were sought by using FACETS based on the information about the total sequence read count and allelic imbalance in endometrial glands and the matched blood samples[90]. Germline polymorphic sites were retrieved from the VCF file generated by the 1000 Genomes Project[77]. The absolute value of the log odds ratio of the variant allele read count in the gland and blood pair was used as the degree of allelic imbalance. After excluding the regions affected by somatic copy number alterations, the mean absolute value of the log odds ratio over the germline heterozygous SNVs in the genome was calculated. Then, the mean was subtracted from each of the absolute value of the log odds ratio to normalize the data to have a mean of zero. Finally, the moving averages of the normalized absolute value of the log odds ratio over 100 heterozygous SNVs were calculated and used for the visualization.

**Reconstruction of phylogenetic trees.** Based on the binary mutation profile according to the presence or absence of somatic SNVs, genetic distances between endometrial glands were computed as the pairwise Hamming distance. The neighbor joining method and the maximum parsimony method were used to reconstruct phylogenetic relationships between the endometrial glands in each subject by using the APE and phangorn packages in the R environment[91–93].

**Identification of putative clonal populations.** The clonal populations present in endometrial glands were explored by clustering somatic SNVs with PyClone[46]. The SNVs were selected by the following criteria: (a) the depth was greater than or equal to 10 in all endometrial glands from a patient; (b) the MAF was greater than or equal to 0.25 in at least one endometrial gland; (c) the mutations did not overlap with the somatic copy number alterations detected by FACETS; and (d) SNVs with low MAF (<0.1) in multiple samples were excluded as putative false positives. In WES, we excluded *TTN*, *MUC3A*, *TAS2R19*, *TAS2R31*, *EDEM2*, *PABPC3*, and *OR8U1* because the SNVs in these genes had low MAF values, and the mutant alleles were found in the matched blood sample, indicating false-positive variant calls. For the mutation sites satisfying these criteria, we compiled MAF profiles of all the mutation sites for all the glands by counting the sequence reads supporting the reference and mutant alleles as described above. The MAF profiles were used for PyClone. Then, clusters with ≥15 mutations were used for subsequent analysis. Then, the results of the SNV clustering together with their MAFs were used as the input to ClonEvol[47]. The polyclonal seeding model was implemented. The number of bootstrap samples was set to 10,000.

**Estimation of timing of genomic events.** Motivated by previous studies[43,44], we estimated the timing of clonal expansions and somatic copy number alterations. The somatic SNVs detected in each group or cluster of glands in a subject were divided into three types: (i) public mutations shared by all glands in a group or cluster, (ii) partially shared mutations shared by a part of the glands in a group or cluster, and (iii) private mutations that were detected only in single glands. Public

mutations are thought to precede the most recent clonal expansion that gave rise to a group or cluster of glands followed by diversifying events, including partially shared mutations and subsequent private mutations. We assumed that the mutation rate for somatic SNVs differed across groups or clusters of glands but remained constant between birth and age at sampling. By using somatic SNVs in regions that were not affected by somatic copy number alterations, the timing of the most recent clonal expansion was estimated as the age at sampling multiplied by the proportion of public mutations to the overall mutational burden in the respective group or cluster of glands as follows:

$$T_{CE} = \text{Age} \times MB_{pub} / \left( MB_{pub} + MB_{ps} + MB_{priv} \right). \quad (6)$$

Let $m = MB_{pub} + MB_{ps} + MB_{priv}$ and $p = MB_{pub} / \left( MB_{pub} + MB_{ps} + MB_{priv} \right)$, 95% CI of $p$ was calculated as: $p \pm 1.96 \times \sqrt{p(1-p)/m}$.

Next, we analyzed the timing of two CN-LOH events at chromosomes 3 and 10 in the endometrium of a subject, which were detected by FACETS. We sought public mutations in the regions affected by these two events. In principle, the MAFs of public mutations that occurred before a CN-LOH are expected to be 1, whereas the MAFs of public mutations that occurred after CN-LOH are expected to be 0.5 when the cellular fraction of the CN-LOH ($\rho$) is 1. We computed the joint probabilities of observing $d_{i,j}^{mut}$ reads supporting the mutant allele in sequence depth $d_{i,j}$ at the $j$ th public SNV site ($j = 1, \ldots, l$) in the $i$ th gland ($i = 1, \ldots, n$) given that the public SNV occurred before ($P_j^{pre}$) or after ($P_j^{post}$) a CN-LOH as follows:

$$P_j^{pre} = \prod_{i=1}^{n} \binom{d_{i,j}}{d_{i,j}^{mut}} \times \rho_i^{d_{i,j}^{mut}} \times \left(1 - \rho_i\right)^{d_{i,j}^{ref}}, \quad (7)$$

and

$$P_j^{post} = \prod_{i=1}^{n} \binom{d_{i,j}}{d_{i,j}^{mut}} \times \left(\frac{\rho_i}{2}\right)^{d_{i,j}^{mut}} \times \left(1 - \frac{\rho_i}{2}\right)^{d_{i,j}^{ref}}, \quad (8)$$

where $d_i^{ref} = d_i - d_i^{mut}$. The value of $\rho_i$ was estimated for each gland by FACETS. Then, the relative ratios of these joint probabilities were used as the weights for the $j$ th public SNV: $w_j^{pre} = P_j^{pre} / \left( P_j^{pre} + P_j^{post} \right)$ and $w_j^{post} = P_j^{post} / \left( P_j^{pre} + P_j^{post} \right)$.

In order to estimate the timing of the CN-LOH and its 95% CI by incorporating the uncertainty of the assignments of the SNVs to before or after the CN-LOH event, we conducted a simple simulation study with 10,000 iterations. By generating random numbers between 0 and 1, we assigned whether each of the variants occurred before or after the CN-LOH event based on the probabilities: $w_j^{pre} = P_j^{pre} / \left( P_j^{pre} + P_j^{post} \right)$ and $w_j^{post} = P_j^{post} / \left( P_j^{pre} + P_j^{post} \right)$. The mutational burdens before and after the CN-LOH, $MB^{pre} = 2 \times$ number of somatic SNVs occurred before the CN-LOH and $MB^{post} =$ number of somatic SNVs occurred after the CN-LOH, respectively. Somatic SNVs that occurred before the CN-LOH resided on the retained allele, and somatic SNVs on the other allele were lost during the CN-LOH. This indicates that the number of somatic SNVs was expected to decrease by half. Therefore, we doubled the number of SNVs occurred before CN-LOH. The proportion $MB_{pre} / \left( MB_{pre} + MB_{post} \right)$ and its variance were calculated for each iteration. Let $m = MB_{pre} + MB_{post}$ and $p = MB_{pre} / \left( MB_{pre} + MB_{post} \right)$, the variance of $p$ was calculated as: $p(1-p)/m$. Next, we estimated the timing of the CN-LOH as follows:

$$T_{CNLOH} = T_{CE} \times MB_{pre} / \left( MB_{pre} + MB_{post} \right). \quad (9)$$

Let $MB_{pub} / \left( MB_{pub} + MB_{ps} + MB_{priv} \right)$ and $MB_{pre} / \left( MB_{pre} + MB_{post} \right)$ be $X$ and $Y$, respectively, the variance of $XY$ can be written as: $V(XY) = V(X)V(Y) + E(X)^2 V(Y) + E(Y)^2 V(X)$. Based on the variance, we calculated the 95% upper and lower limits of $T_{CNLOH}$ in each iteration. After the simulation study, the means over the 10,000 iterations were calculated as the estimate for $T_{CNLOH}$ and its 95% CI.

**Visualization for statistical analyses and distribution of somatic mutations.** The R packages, ggplot2 (Version 3.3.3), ggjoy (Version 0.4.1) and ggridges (Version 0.5.3), were used to visualize the results of statistical analyses. The mutational spectrum was depicted by the barplot3d R package (Version 1.0.1). The lolipop plots displaying mutation distribution and protein domains were created by the trackViewer R package (Version 1.22.1).

**Whole-mount 3D staining of endometrial tissue.** For the whole-mount 3D staining of human endometrial tissue, we used the updated clear, unobstructed brain/body imaging cocktails and computational analysis (CUBIC) protocols described in our previous studies[31,45]. Briefly, endometrial blocks (55 to 635 mm³) were stored in formalin until use. The tissue blocks were washed with PBS for 6 h before clearing. Then, the tissue blocks were immersed in CUBIC-L [T3740 (mixture of 10 wt% N-butyldiethanolamine and 10 wt% Triton X-100), Tokyo

Chemical Industry] with shaking at 45 °C for 6 days. During delipidation, the CUBIC-L was refreshed once. After the samples were washed with PBS for several hours, the tissue blocks were placed in 1 ml of immunostaining buffer (mixture of PBS, 0.5% Triton X-100, 0.25% casein, and 0.01% NaN₃) containing 1:100 diluted Alexa 555-conjugated CK7 antibody (ab203434, Abcam) for 14 days at room temperature with gentle shaking. After washing again with PBS for several hours, the samples were then subjected to postfixation by 1% PFA in 0.1 M PB at room temperature for 5 hours with gentle shaking. The tissue samples were immersed in 1:1 diluted CUBIC-R+ [T3741 (mixture of 45 wt% 2,3-dimethyl-1-phenyl-5-pryrazolone, 30 wt% nicotinamide and 5 wt% N-butyldiethanolamine), Tokyo Chemical Industry] with gentle shaking at room temperature for 1 day. The tissue samples were then immersed in CUBIC-R+ with gentle shaking at room temperature for 2 days.

**Microscopy**. Macroscopic whole-mount images were acquired with a light-sheet fluorescence (LSF) microscope (MVX10-LS, Olympus) using a ×0.63 objective lens [numerical aperture = 0.15, working distance = 87 mm] with digital zoom ×3.2. Voxel resolution was set as follows: x = 3.27 µm, y = 3.27 µm, and z = 5.0 µm. The LSF microscope was equipped with lasers emitting at 488 nm and 532 nm. When the stage was moved in the axial direction, the detection objective lens was synchronically moved to the axial direction to prevent defocusing. The Alexa555 signals of CK7-expressing endometrial epithelial cells were measured by excitation with 532 nm lasers. Autofluorescence was measured by excitation at 488 nm.

**3D image analysis**. All raw image data were collected in lossless 16-bit TIFF format. All CK7 fluorescence images were obtained by subtracting the background and applying an unsharp mask using Fiji software[94]. Three-dimensionally rendered images were visualized, captured and analyzed with Imaris software (Version 9.5.1, Bitplane). The image analysis by Imaris software was performed as described in our previous study[31]. Briefly, TIFF files were imported in the Surpass mode of Imaris. The reconstituted 3D images were cropped to a region of interest using the 3D Crop function. Using the channel arithmetic function, the CK7 signal was removed from the autofluorescence signal to create a channel with only endometrial epithelium and gland signals. To identify glandular continuity, we observed the shapes of glands using continuous tomographic images from the XY, XZ, and YZ planes. Then, we selected one unit of the glands sharing the rhizome. The glands sharing the rhizome were traced manually, and their 3D structures were reconstructed by the Surface module. The 3D surface object was pseudocolored and separated into a new channel. Thus, the glands sharing the rhizome were visualized. To calculate the surface area of the endometrium occupied by the openings of the glands sharing a rhizome, the XYZ coordinates of the tips of the glands of the 3D surface objects were measured using the Measurement Point module. We formed triangles by selecting three points of the tips of the glands so that the resulting triangles did not overlap with each other. Then, the surface area was calculated as the sum of the areas of the triangles. Among the surface areas for all possible combinations of the triangles, the smallest one was selected. Because subject No. 40 had a large number of glands sharing the rhizome, six glands located on the outside edge were selected. The snapshot and animation functions were used to capture images and videos, respectively.

**3D modeling of laser-microdissected glands**. To create the 3D model of laser microdissected glands, 62 serial cryosection images before laser microdissection were imported into Photoshop 2020 (Adobe). The first image in the image stack was selected as a reference image and used to align subsequent images. Thus, all images were aligned to their neighbors. The shapes of endometrial glands were drawn manually in new layers with a transparent background on each 2D serial cryosection image, which were exported individually as TIFF images. Three-dimensionally rendered images were visualized and captured with Imaris software (Version 9.5.1, Bitplane) as mentioned above. Briefly, TIFF files were imported in the Surpass mode of Imaris. Using the channel arithmetic function, the signals of the gland depicted were distinguishable from the background and used for 3D reconstruction by the Surface module. The snapshot and animation functions were used to capture images and videos, respectively.

**Reporting summary**. Further information on research design is available in the Nature Research Reporting Summary linked to this article.

## Data availability

The raw target-gene sequencing data and WES/WGS data have been deposited in the European Genome-Phenome Archive (http://www.ebi.ac.uk/ega/) under the accession numbers EGAS00001005914 and EGAS00001005822, respectively. These sequencing data are available under restricted access. In order to obtain these data, researchers need to contact the main member of the Data Access Committee (DAC) (K. Yoshihara) at yoshikou@med.niigata-u.ac.jp and make scientifically appropriate requests. All requests will be reviewed by the institutional review board (IRB) of the Niigata University, and will require the requesting researcher to sign a data access agreement with the Niigata University. The raw and processed imaging datasets acquired by LMD7 laser microdissection microscope and 3D images are available at https://true.med.niigata-u.ac.jp/ncomm2021/. Source data are provided with this paper.

## Code availability

Code for statistical analyses on mutation burden, mutational signatures, and timing of genome events including clonal expansions and copy neutral loss-of-heterozygosity is deposited on GitHub at https://github.com/HirofumiNakaoka/endometrium_natcommun_2021.

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

## Acknowledgements

This work was supported in part by the Japan Society for the Promotion of Science (JSPS) KAKENHI grant numbers 17H04336 (Grant-in-Aid for Scientific Research B to T.E.), 19K09822 (Grant-in-Aid for Scientific Research C to K. Yoshihara), 17K08600 (Grant-in-Aid for Scientific Research C to H.N.), 20K07318 (Grant-in-Aid for Scientific Research C to H.N.) and 16H06279 (Grant-in-Aid for Scientific Research on Innovative Areas Platforms for Advanced Technologies and Research Resources to H.N.). This work was supported by a Challenging Exploratory Research Projects for the Future grant to H.N.

from the Research Organization of Information and Systems (ROIS) and a Research Grant to K. Yoshihara from The Uehara Memorial Foundation. We are grateful to Anna Ishida, Kenji Ohyachi, Junko Kajiwara, Junko Kitayama, Yumiko Sato, and Keiko Nishikawa for their technical assistance.

## Author contributions

M.Y., H.N., K. Suda and K. Saito performed experiments. M.Y., H.N., K. Yoshihara, and S.R. performed data analyses. M.Y., K. Suda, T.I., N.Y., H.U., K. Sugino, Y.M., K. Yamawaki, and R.T. collected endometrial samples. T.M. performed histological validation. M.Y., H.N., K. Suda, K. Yoshihara, and T.E. designed the project. H.N. and K. Yoshihara contributed to the project coordination. K.T., R.V. I.I., and T.E. supervised and supported the project. M.Y., H.N., and K.Y. wrote the manuscript. All authors reviewed and approved the final manuscript.

## Competing interests

K.T. is a co-inventor on patent applications covering the CUBIC reagents (PCT/JP2014/070618 [pending], "Composition for preparing biomaterial with excellent light-transmitting property, and use thereof", patent applicant is RIKEN; PCT/JP2017/016410 [pending], "Composition for preparing biological material having excellent light transmissivity and use of composition", patent applicant is RIKEN). The remaining authors declare no competing interests.
