## [Peer review file · Nature Communications]

The rebuttal to Reviewer #2 refers to Yamaguchi et al., iScience 2021. Yamaguchi M, Yoshihara K, Suda K, Nakaoka H, Yachida N, Ueda H, Sugino K, Mori Y, Yamawaki K, Tamura R, Ishiguro T, Motoyama T, Watanabe Y, Okuda S, Tainaka K, Enomoto T. Three-dimensional understanding of the morphological complexity of the human uterine endometrium. iScience 2;24(4):102258 (2021). doi: 10.1016/j.isci.2021.102258.

REVIEWER COMMENTS

Reviewer #1 (Remarks to the Author):

The authors have performed a deep study of mutational landscape and clonal dynamics in normal endometrial tissues across multiple individuals. They have done combination of targeted, whole exome and genome sequencing on spatially isolated endometrium glands. They showed an age and menstrual cycle effect on mutation burden in this tissue. Moreover, they have reported three main mutational signatures are operative in this tissue namely: SBS1, SBS5 and SBS18. Driver landscape analysis showed clear positive cancer gene selection in this tissue in line with previous publications. They identified regional/chromosomal arm CN-LOH events which may have selection advantages in normal endometrium glands and these events can be tolerated years before cancer progression. Overall, the work is robust and most of the findings confirms the recent published works in this field.

Below are my main comments:

1- I couldn't find information about the mutation burden analysis; can the authors expand on their mutation model? Have they tested other effect that may affect mutation burden other than age, and CNMCs? For endometrium glands, it is extremely important to test the effect of: BMI, age of puberty and menopause, parity on both mutation rate and driver.

2- To increase their statistical power to test the effect of some the phenotypes mentioned above, I suggest the authors to download and merged data from recently published data on normal endometrium. This may lead to some novel findings which were not possible to address previously due to small sample size.

3- For the mutational signature analysis, the authors have identified three main signatures which were also reported previously. Do they extract signatures which may correlate with different pathology e.g. in cases with ovarian dermoid or adenomyosis? doing a signature extraction across all individuals together may preclude extraction of specific signature which may be present in a small subset of samples. Hence, they need to do signature extraction per each sub-group separately.

4- What is the age effect on driver landscape? Do the authors identified specific driver genes associating with certain clinical assertion such as ovarian endometriosis?

5- They showed CN-LOH events in normal glands which is in line with some of the recent publications. Other than CN-LOH events, what is the indel, structural variation and copy number changes in endometrium with age?

6- Despite driver gene analysis, I couldn't find the details about clonality across glands? Is there any relationship between clonality and sequencing depth? How many of the glands showed sub-clonality?

Minor comments:

1- For variant allele fraction analysis of SNVs, were they corrected for the copy number? I couldn't find this in the method section.

2- Does the extent of clonal expansion correlate with prognosis?

3- Description of some of the statistical testes are missing in the main text. Please add this information.

Reviewer #2 (Remarks to the Author):

This is quite an interesting paper. It follows a recent series of WES and WGS studies performed on the glandular epithelial cells of the human endometrium.

1- The discovery regarding the clonality of epithelial cells in glandular crypts came out in the papers of Moore, Inoue and Suda. None of these previous papers however emphasized the rhizome structure. The authors of the current manuscript did a more thorough and complete job of accurately characterizing the mutational landscape. They need to acknowledge and cite the Inoue paper though (PMID: 31857578). Although the focus is adenomyosis, this paper contains a lot of discussion of about clonality.

2- The Tempest paper published in 2020 focuses on the "complex horizontally interconnecting network of basalis glands" (PMID: 32476144). However, this group did not base this model on a sophisticated and thorough mutational analysis as in the current paper. The authors need to acknowledge that the rhizome story has been proposed at least by another group. They cited this paper but did not fully acknowledge what was revealed in it.

3- The authors should rewrite the abstract to reflect the authenticity or novelty of their discovery and to distinguish this from previously published papers

4- To add some more novelty, I suggest that the authors check the expression of previously published endometrial epithelial progenitor cell markers by the Hapangama and Gargett (an possibly other) groups and correlate these with distinct mutations.

5- One particularly interesting question is, "how does each mutation biologically affect the progenitor/stem nature of the epithelial cell in the glandular crypt or rhizome?" Also what are the spatial relationships of this rhizome structure with the blood vessel network? Perhaps the authors could add these to the discussion.

Reviewer #3 (Remarks to the Author):

The authors describe an extensive dataset of normal endometrial epithelium and attempt to answer important and topical questions surrounding somatic mutation in normal tissues. The results are compelling and overall the analyses are well constructed and the paper well written. I have the following major and minor concerns:

Major:

1. The authors did not include SBS3, the signature of homologous recombination deficiency, despite its high prevalence in ovarian cancer.
2. BRCA status is notably absent and would be an important clinical covariate. Why was it not assessed?
3. The authors state: "A total of 29,000,000 somatic SNVs in noncoding regions from the COSMIC database were classified into 96 mutation..." I couldnt figure out which cosmic dataset was used. The table of non-coding variants available here: <https://cancer.sanger.ac.uk/cosmic/download> appear to have only 20M variants. Also, are the authors estimating substitution rates in trinucleotide contexts based on variants from all cancer types? Restricting to cervical cancer, endometrial cancer and ovarian cancer as was done for the signatures would seem more appropriate.
4. The authors should explain the motivation for the criteria: "SNVs with low MAF (<0.1) in multiple samples were excluded". Such variants would either be artifacts or, less likely, evidence of clones that populate multiple samples. It would be helpful to find a better argument for removing them than the fact that they represent a biologically unlikely scenario.
5. Are the authors able to rule out clonal hematopoiesis as a source of variants in older patients?
6. For the estimates of the timings of CN-LOH and clonal expansions, please provide a measure of uncertainty, possibly as a confidence interval.
7. For subject 35, why were the anterior and posterior walls analyzed independently? The authors conclude independent FGFR2 mutated at a hotspot but this seems to require more explanation and

justification. Why is the non-contiguous distribution of FGFR2 S252W not the result of the rhizome structure?

Minor:

1. This sentence is somewhat confusing: "We used endometrial glands whose coverages of the target region of at least 20 reads were greater than 0.70 for subsequent analyses."
2. This sentence could be written more clearly: "We identified significant SBS signatures if the lower end of the 90% highest posterior density interval for the estimated exposure was greater than the threshold value (0.01)."
3. What form of multiple test correction was used for identifying significant positively selected genes?

We are grateful to the editor and three reviewers for the critical and valuable suggestions that helped us to improve our paper considerably. As indicated in the responses below, we have taken all these comments and suggestions into considerations.

Response to the comments by Reviewer #1.

General Comment.

The authors have performed a deep study of mutational landscape and clonal dynamics in normal endometrial tissues across multiple individuals. They have done combination of targeted, whole exome and genome sequencing on spatially isolated endometrium glands. They showed an age and menstrual cycle effect on mutation burden in this tissue. Moreover, they have reported three main mutational signatures are operative in this tissue namely: SBS1, SBS5 and SBS18. Driver landscape analysis showed clear positive cancer gene selection in this tissue in line with previous publications. They identified regional/chromosomal arm copy neutral loss-of-heterozygosity (CN-LOH) events which may have selection advantages in normal endometrium glands and these events can be tolerated years before cancer progression. Overall, the work is robust and most of the findings confirms the recent published works in this field.

Response.

We appreciate the reviewer's positive evaluation on our manuscript. As the reviewer pointed out, our results corroborate previous findings such as age-related accumulation of somatic mutations, strong positive selections acting on cancer-related gene mutations and mutational signatures in normal human endometrial glands.

At the same time, the results of our study provide novel findings. We showed that endometrial glands derived from the same mutant clones occupied substantial areas of the endometrium (Figs 4 & 5). To clarify the mechanism underlying this finding, we focused on the glandular structure at the bottom of the endometrium ran horizontally along the muscular layer (rhizome structure), and several branches rose from the rhizome structure toward the luminal epithelium (Fig 6). We demonstrated that multiple glands that were linked through a rhizome structure had a shared clonal origin (Fig 7). Our results suggest that clonal expansions through rhizome structures are involved in the

mechanism by which mutant clones extend their territories in the endometrium.

Major Comment #1.

I couldn't find information about the mutation burden analysis; can the authors expand on their mutation model? Have they tested other effect that may affect mutation burden other than age, and cumulative number of menstrual cycles (CNMCs)? For endometrium glands, it is extremely important to test the effect of: BMI, age of puberty and menopause, parity on both mutation rate and driver.

Response.

We analyzed the effects of age, CNMCs, age of menarche which is one of the best indicators for age of puberty, parity, BMI, pack years of smoking, and disease status on the burden of somatic single nucleotide variants (SNVs) with high mutant allele frequencies (MAFs). As shown in Fig 2, age and CNMCs were strongly correlated with the mutational burden ($P < 0.0001$), and age of menarche showed a suggestive association with the mutational burden ($P = 0.09$). The other clinical variables were not significantly correlated with the mutational burden (Supplementary Figure 2).

The study participants' ages ranged from 21 to 53 years, and there were no postmenopausal women. This is because that the endometrial tissue becomes atrophic after menopause and therefore it is difficult to isolate intact single endometrial glands from atrophic endometrium in postmenopausal women. According to the reviewer's comment, we examined the associations between clinical variables and the burden of mutations in cancer-driver genes (*ARID1A*, *CTNNB1*, *FBXW7*, *KRAS*, *PIK3CA*, *PIK3R1*, *PPP2R1A*, *PTEN*, and *TP53*). Age and CNMCs were positively correlated with the burden of driver mutations ($P < 0.0001$), and age of menarche showed a significant negative correlation after the adjustment for age ($P = 0.0034$) (Supplementary Figure 3).

We rewrote these results more explicitly in the main text. Additionally, the definition of cancer-driver gene mutations is described in the Methods section.

Major Comment #2.

To increase their statistical power to test the effect of some the phenotypes mentioned above, I suggest the authors to download and merged data from recently published data on normal endometrium. This may lead to some novel findings which were not possible to address previously due to small sample size.

Response.

Thank you for your valuable suggestion. We searched Pubmed and found three recently published studies on somatic mutations in normal endometrium (Supporting Table 1-1). Of those studies, only Moore et al. seems to deposit sequencing data for purified normal endometrial epithelium samples in public database and contain detailed clinical information other than age. Therefore, we considered the data by Moore et al. for pooled analysis. The data by Moore et al. is controlled access data deposited in European Genome-Phenome Archive (accession number, EGAS00001002471). On July 1st (soon after we received this decision letter), we contacted the Sanger Institute to use EGAS00001002471. Unfortunately, we have not received the approval for access to the dataset. In addition, we asked the data manager of the Sanger Institute about the use of clinical information per each sample on July 9th, but we have not received any answers yet.

Here, we would like to discuss the comparability of Moore's dataset (number of women, $n = 28$) and our dataset ($n = 32$). While Moore et al. conducted whole-genome sequencing, we used the dataset based on target sequencing for 112 genes to estimate the mutational burden. We selected these 112 genes that were frequently mutated in endometrial and ovarian cancers (Lawrence et al. Nature 2014; Jones et al. Science 2010) and in endometriosis and normal endometrium (Suda et al. Cell Rep 2018). In addition, the average depth of our target sequence (87.7) was much higher than that of whole-genome sequence data (28). The mutational burdens calculated based on deep target sequencing for exons of highly mutated genes may differ from those calculated based on whole-genome sequencing. At the same time, distributions of clinical variables of participants differ between the two studies. We retrieved the information about clinical variables for the subjects from Supplementary Result 1 of the previous study (Moore et al. Nature 2020). As shown in Supporting Figure 1-1, the averages of age and BMI were different between the two studies at suggestive and nominal significance levels ($P = 0.055$ and $P = 0.031$, respectively) by Welch's t-test. Therefore, the distributions of mutational burdens and clinical variables differ between these two studies. If we examine correlations between the mutational burdens and clinical variables, there is a possibility that we might detect spurious associations due to systematic differences in these two datasets. We would like to conduct the pooled analysis suggested by the reviewer after

increasing whole-genome sequence data in future.

Study (Year of publication)	Number of sequencing subjects	Enrichment of the endometrial epithelial cells	Sequencing method	Availability of sequence data	Information about clinical variables for the subjects
Lac et al. J Pathol (2019)	110	Yes (manually macrodissection)	Target gene sequencing	None	Age, Menstrual phase
Moore et al. Nature (2020)	28	Yes (laser microdissection)	WGS	Available	Age, BMI, Parity, Menstrual phase, Menopause status
Aguilar et al. J Pathol (2021)	19	No	Target gene sequencing	Available	Age
Yamaguchi et al. (current study)	32 (for the analysis of mutation burden)	Yes (isolated single glands by collagenase digestion)	Target gene sequencing (for the analysis of mutation burden)	Available	Age, BMI, Parity, Menstrual cycle (days), Menarche (age), CNMC, Pack-years

Supporting Table 1-1. List of the recently published studies on somatic mutations in normal endometrium

CNMC: cumulative number of menstrual cycles

Supporting Figure 1-1. Comparison of clinical information on the subjects of Yamaguchi et al. and Moore et al.

Left panels: Density plots. Right panels: Box plots show the median and interquartile range (IQR), with whiskers indicating the 1.5 IQR. Data were statistically compared by t test.

Major Comment #3.

For the mutational signature analysis, the authors have identified three main signatures which were also reported previously. Do they extract signatures which may correlate with different pathology e.g. in cases with ovarian dermoid or adenomyosis? doing a signature extraction across all individuals together may preclude extraction of specific signature which may be present in a small subset of samples. Hence, they need to do signature extraction per each subgroup separately.

Response.

Thank you for your constructive suggestion. We conducted signature extraction by classifying patients into four disease subgroups: i) cervical neoplasia, ii) dermoid cyst, iii) endometrium-related diseases including endometriosis and adenomyosis, and iv) myoma. Similar to the result analyzing all patients together, we detected SBS1, SBS5 and SBS18 in the subgroup analyses as shown in

Supporting Figure 1-2.

Supporting Figure 1-2.

Bar chart representing the results of Bayesian inference by using sigfit to determine the contribution of the COSMIC mutational signatures to somatic SNVs with high MAF for each of the four disease subgroups. Blue bars showing significant mutational signatures. The inset is a doughnut chart summarizing the contributions of significant mutational signatures.

Major Comment #4.

What is the age effect on diver landscape? Do the authors identified specific driver genes associating with certain clinical assertion such as ovarian endometriosis?

Response.

As described in the response to Major Comment #1, age and CNMCs were positively correlated with the burden of driver mutations ($P < 0.0001$), and age of

menarche showed a significant negative correlation after the adjustment for age (P = 0.0034) (Supplementary Figure 3).

The average age and CNMCs of patients with *KRAS* mutations exhibiting allelic imbalance (i.e., mutant allele frequency ≥ 0.8) were higher (P = 0.041 and P = 0.021, respectively) by Wilcoxon rank-sum test. Additionally, *KRAS* mutations exhibiting allelic imbalance were overrepresented in patients with endometriosis (P = 0.012) by Fisher's exact test.

We added these results in the main text.

Major Comment #5.

They showed CN-LOH events in normal glands which is in line with some of the recent publications. Other than CN-LOH events, what is the indel, structural variation and copy number changes in endometrium with age?

Response.

As described in the response to Major Comment #4, we identified significant associations between *KRAS* mutations with allelic imbalance (signatures of copy number alterations or CN-LOH events) and clinical variables. We could not detect structural variants in this study. We assessed whether the burden of short insertions and deletions (indels) was correlated with clinical variables; however, there was no significant association as shown in Supporting Figure 1-3.

Supporting Figure 1-3.

a) Bar charts showing the burdens of somatic INDELs with high MAF for 32 subjects. The burden is defined as the number of high-MAF INDELs per Mbp sequenced. Multiple glands from an individual subject are pooled. Subjects are sorted in ascending order according to mutation burden. The heatmap below the bar charts represents age, CNMCs, and pack-years of cigarette smoking for the 32 subjects.

b) Linear relationships of clinical variables with the burden of somatic INDELs with high MAF.

c) Relationships of the affected status of gynecologic cancers and benign diseases, sampling methods and smoking status with the burden of somatic INDELs with high MAF.

Major Comment #6.

Despite driver gene analysis, I couldn't find the details about clonality across glands? Is there any relationship between clonality and sequencing depth? How many of the glands showed sub-clonality?

Response.

In this study, we evaluated clonality of each endometrial gland based on mutant allele frequencies. According to the density plot for MAFs of somatic SNVs (Fig 1e), we defined mutations whose MAFs were greater than or equal to 0.25 as “high MAF” mutations. The high MAF mutations are the indicator for the presence of a major mutant clone within the gland. The low MAF mutations (MAF < 0.25) suggests the presence of subclonal cell populations within glands (Fig 1e). 61.8% (=551/891) and 50.1% (=446/891) of glands harbored high and low MAF mutations, respectively. We added this information in the main text.

In addition, we evaluated clonality across endometrial glands by the presence of multiple shared high MAF mutations. The inter-gland clonality analysis is our main focus and provides novel findings for the mechanism of clonal expansions through rhizome structures in the endometrium (Figs 4, 5, and 7).

We purified epithelial compartments of the endometrium by using single gland isolation and laser-microdissection. Therefore, we think that the sequencing depth is sufficient to detect high MAF mutations. If we focus more on rare cell populations (e.g., mutant clones accounting for 1% of epithelial cells in an endometrial gland), we need to increase sequence depth.

Minor Comment #1.

For variant allele fraction analysis of SNVs, were they corrected for the copy number? I couldn't find this in the method section.

Response.

Cancer cell fractions are estimated by considering the copy number status and used to cluster mutations and to determine the order of mutational events to reconstruct evolutionary relationships among mutant clones (Nik-Zainal et al. Cell 2012). In our study, we used whole-exome or whole-genome sequence data to infer clonal relationships among endometrial glands (Figures 4, 5 and 7). Somatic copy number alterations were very rare in the endometrial glands analyzed in this study. Therefore, we simply excluded the regions affected by somatic copy number alterations from the analyses for reconstructing phylogenetic trees. This is described in the subsection “Identification of putative clonal populations” of the Methods.

Minor Comment #2.

Does the extend of clonal expansion correlate with prognosis?

Response.

We speculate that the accumulation of somatic mutations in cancer-associated genes and their extensive clonal expansion in the endometrial epithelium may be associated with the development of endometrium-related diseases, but long-term follow-up is needed to confirm the prognosis of individuals.

According to the reviewer's comment, we added the sentence to Discussion section as follows (page 26): As clonal hematopoiesis increases the risks for blood cancer and cardiovascular disease, there is a possibility that the overrepresentation of endometrial glands derived from a single clone raise the risks for endometrium-related diseases. Long-term follow-up studies are needed to determine whether mutant clones with cancer-associated gene mutations extends their territories over time and whether the situation where the of endometrial epithelial cells is dominated by a small number of mutant clones increases the risks for endometrium-related diseases such as adenomyosis, endometriosis and endometrial cancer.

Minor Comment #3.

Description of some of the statistical testes are missing in the main text. Please add this information.

Response.

Thank you for pointing out that the descriptions of some of the statistical tests are not sufficient. We checked the manuscript and added the descriptions.

Response to the comments by Reviewer #2.

General Comment.

This is quite an interesting paper. It follows a recent series of WES and WGS studies performed on the glandular epithelial cells of the human endometrium.

Response.

We deeply appreciate the reviewer's positive evaluation on our manuscript.

Major Comments #1.

The discovery regarding the clonality of epithelial cells in glandular crypts came out in the papers of Moore, Inoue and Suda. None of these previous papers however emphasized the rhizome structure. The authors of the current manuscript did a more thorough and complete job of accurately characterizing the mutational landscape. They need to acknowledge and cite the Inoue paper though (PMID: 31857578). Although the focus is adenomyosis, this paper contains a lot of discussion of about clonality.

Response.

According to the reviewer's comment, we cited the paper by Inoue et al. (PMID: 31857578) as follows (page 25):

A recent study showed that adenomyosis and histologically normal endometrium adjacent to the adenomyotic lesions had identical *KRAS* hotspot mutations, suggesting that *KRAS*-mutated adenomyotic clones originate from normal endometrium.

Major Comment #2.

The Tempest paper published in 2020 focuses on the "complex horizontally interconnecting network of basalis glands" (PMID: 32476144). However, this group did not base this model on a sophisticated and thorough mutational analysis as in the current paper. The authors need to acknowledge that the rhizome story has been proposed at least by another group. They cited this paper but did not fully acknowledge what was revealed in it.

Response.

In the original version of our manuscript, we mentioned that Tempest et al.

discovered the complex horizontally interconnecting network of basalis glands in the Introduction, Results and Discussion. Additionally, in the Discussion, we mentioned that Tempest et al. showed the contribution of multiple stem-cell populations to the regeneration of single endometrial glands by using somatic mutations in mitochondrial DNA.

According to the reviewer's comment, we added their inference about functional roles of rhizome-structures as follows (page 23):

Tempest et al. have proposed that rhizome-structures assist self-preservation, self-renewal, and scarless regeneration of the human endometrium as a niche of endometrial epithelial stem/progenitor cells.

We added a citation of Tempest et al. to the following sentence (page 24). The presence of rhizome structures might be beneficial for post-menstrual endometrial repair by protecting the endometrial stem/progenitor cells from shedding at the menstrual phase.

Major Comment #3.

The authors should rewrite the abstract to reflect the authenticity or novelty of their discovery and to distinguish this from previously published papers

Response.

We revised the Abstract. In the revised version of the Abstract, we did not include our results validating previous findings. We focused only on our novel findings and added the novel result regarding the chronological ages at which clonal expansions and copy-neutral loss-of-function events occurred.

Major Comment #4.

To add some more novelty, I suggest that the authors check the expression of previously published endometrial epithelial progenitor cell markers by the Hapangama and Gargett (an possibly other) groups and correlate these with distinct mutations.

Response.

We appreciate your suggestion. We examined expressions of previously reported endometrial epithelial progenitor cell markers (SSEA-1 [Valentijn et al. Hum Reprod. 2013] and LGR5 [Tempest et al. Hum Reprod. 2018]) in endometrial glandular epithelium by using RNA in situ hybridization with RNA

scope technology (Wang et al. J Mol Diagn. 2012). As a result, we confirmed that the expression patterns of these two markers were consistent with previous reports (Supporting Figure 2-1). Unfortunately, we have not been able to develop the methods to accurately measure expression levels of these markers in each gland and to quantitatively evaluate whether there are differences in expression levels of these markers among glands within the same case. At the same time, we have not been able to establish the method to simultaneously evaluate mutation status and expression levels of these markers in each gland. After solving these technical issues, we would like to elucidate the relationship between mutation status and expression levels of progenitor cell markers in endometrial glands.

Supporting Figure 2-1. Expressions of SSEA-1 and LGR5 in proliferative human endometrium by RNA scope.

Left panel: SSEA-1 expression in the basalis. Right panel: LGR5 expression in the luminal epithelium.

Major Comment #5.

One particularly interesting question is, “how does each mutation biologically affect the progenitor/stem nature of the epithelial cell in the glandular crypt or rhizome?” Also what are the spatial relationships of this rhizome structure with the blood vessel network? Perhaps the authors could add these to the discussion.

Response.

We thank Reviewer #2 for the insightful comment. We speculate that the rhizome structure is the niche of endometrial epithelial stem cells, and that clones with cancer-related gene mutations may confer a proliferative advantage and contribute to tissue regeneration. We also have a major interest in the spatial relationships of the rhizome structure with the blood vessel network. By performing 3D multicolor fluorescence imaging with CK7 and cluster of differentiation (CD)31 in adenomyosis samples, we revealed that part of the ectopic endometrial glands expanded along the route of large blood vessels in myometrium (Yamaguchi et al, iScience 2021) as shown in Supporting Figure 2-2.

Supporting Figure 2-2. 3D multicolor fluorescence imaging of adenomyotic tissue with anti-CK7 and anti-CD31 antibodies (Yamaguchi et al. iScience 2021). Yellow object: ectopic endometrial glands in the myometrium.

Yellow objects were output by the Surface module in Imaris software based on the CK7 signal.

Red arrows indicate where ectopic glands extend along large blood vessels.

Autofluo, autofluorescence

We have been working on establishing the method for 3D co-staining of glandular epithelium and blood vessels in the endometrium, but it has some technical problems to be solved. In the future, we would like to examine the spatial relationship between rhizome and vascular travel in order to solve the mystery of how rhizome structures are formed and spread.

According to the reviewer's comment, we added two sentences highlighted in yellow to Discussion section as follows (page 25):

The mechanisms of rhizome structure formation in the human endometrium are totally unknown. Anatomical, embryological and physiological studies are required to elucidate when and how the rhizome structures develop. For this purpose, it will be helpful to clarify the spatial relationship between rhizome structure and vascular network by 3D imaging. The biological and medical significance of the rhizome structures also remains obscure. We speculate that the rhizome structures act as a double-edged sword. The presence of the rhizome structures might be beneficial for post-menstrual endometrial repair by protecting the endometrial stem/progenitor cells from shedding at the menstrual phase. We presume that clones with cancer-associated gene mutations may confer a proliferative advantage and contribute to stable tissue regeneration by expanding the area of rhizome structure.

Response to the comments by Reviewer #3.

General Comment.

The authors describe an extensive dataset of normal endometrial epithelium and attempt to answer important and topical questions surrounding somatic mutation in normal tissues. The results are compelling and overall the analyses are well constructed and the paper well written. I have the following major and minor concerns:

Response.

We appreciate that reviewer #3 provided valuable comments to improve our manuscript.

Major Comment #1.

The authors did not include SBS3, the signature of homologous recombination deficiency, despite its high prevalence in ovarian cancer.

Response.

Thank you for pointing out the important issue. According to the reviewer's comment, we examined mutational signatures including SBS3. As the result, SBS3 was not significant. We have reflected the result of the reanalysis in Figure 1, panel g and added descriptions in the Methods section.

Major Comment #2.

BRCA status is notably absent and would be an important clinical covariate. Why was it not assessed?

Response.

We have included BRCA1 and BRCA2 in our target gene sequencing panel as shown in Supplementary Table 2. Only 0.2% (2/891) of single gland samples harbored BRCA mutations as shown in Supporting Table 3-1. Therefore, we did not mention them in the main text.

sample	chr	pos	ref	alt	gene	effect	hgvs_p	maf
S31-gland24	chr13	32329469	G	C	BRCA2	missense_variant	NP_000050.2:p.Val220Leu	0.387
S26-gland35	chr13	32336718	G	T	BRCA2	missense_variant	NP_000050.2:p.Gly788Val	0.068

Supporting Table 3-1. BRCA mutations detected in the target gene sequencing for 891 endometrial glands from 32 women.

Major Comment #3.

The authors state: “A total of ~29,000,000 somatic SNVs in noncoding regions from the COSMIC database were classified into 96 mutation...” I couldnt figure out which cosmic dataset was used. The table of non-coding variants available here: <https://cancer.sanger.ac.uk/cosmic/download> appear to have only ~20M variants. Also, are the authors estimating substitution rates in trinucleotide contexts based on variants from all cancer types? Restricting to cervical cancer, endometrial cancer and ovarian cancer as was done for the signatures would seem more appropriate.

Response.

Thank you for pointing out that our description about the noncoding variants from the COSMIC database was not appropriate. We used the file “CosmicNonCodingVariants.vcf” of v90 (September 2019). This file contains “CNT” tag that provides information about how many samples have the same mutation. By considering the “CNT” information, we counted mutational events (i.e., if a variant is identified in two samples, we count two mutational events). In total, about 29 million SNV events are included. In this file, the same mutations are recorded in multiple lines, if the mutation has multiple different genetic annotations. After excluding the redundant records and polymorphic SNVs, about 14 million somatic SNV events were available. In the original version of our analysis, we used the 14 million somatic SNV events. We should have written this value (about 14 million); however, we mistakenly wrote the value before filtering (about 29 million). This is because that we did not fully describe the filtering steps.

According to the reviewer’s comment, we reanalyzed selections of somatic mutations in the normal endometrium by using noncoding SNVs detected in cervical cancer, endometrial cancer and ovarian cancer. The file “CosmicNonCodingVariants.vcf” does not contain the information about the primary tissue from which samples originated. Therefore, we used the file “CosmicNCV.tsv” of v94 (May 2021) that contains the information about the primary tissue. First, we retrieved non-coding somatic SNVs by using the following filtering criteria: i) inclusion of SNVs confirmed to be somatic, and ii)

exclusion of known single nucleotide polymorphisms. We obtained about 15 million SNV events in all cancer types. Then, we selected SNVs if the primary site was cervix, endometrium or ovary. As a consequence, 1,136,781 SNV events in the three cancer types were retrieved and used for estimating substitution rates in trinucleotide contexts. The results of positively selected genes did not change between the previous and current analyses. We revised the Results and Methods sections and Figure 3 (panels g-i) accordingly.

Major Comment #4.

The authors should explain the motivation for the criteria: “SNVs with low MAF (<0.1) in multiple samples were excluded”. Such variants would either be artifacts or, less likely, evidence of clones that populate multiple samples. It would be helpful to find a better argument for removing them than the fact that they represent a biologically unlikely scenario.

Response.

As the reviewer considered, we think that SNVs with low MAF (<0.1) in multiple samples are likely to be artifacts. The mutant alleles of such SNVs are sometimes detected in blood-derived control samples probably due to misalignment of reads. These properties for suspected false positive variant calls are commonly seen in TTN and MUC. Therefore, we used this criterion to exclude false positive mutation calls with similar patterns.

According to the reviewer’s comment, we re-structured the sentence as follows (page 43):

(d) SNVs with low MAF (<0.1) in multiple samples, a characteristic of suspected false-positive mutation calls in genes such as *TTN* and mucin family genes probably due to misalignment of reads, were excluded.

Major Comment #5.

Are the authors able to rule out clonal hematopoiesis as a source of variants in older patients?

Response.

We detected somatic mutations in endometrial epithelial cells as the DNA sequences that were different from the human reference genome sequence. To avoid misclassification between somatic and germline variants in each individual,

blood-derived DNA samples from the same individual was used as a control. If mutations derived from clonal hematopoiesis are present in blood-derived DNA of older women, genomes of endometrial epithelial cells are not changed. Therefore, mutations derived from clonal hematopoiesis do not affect the detection of somatic mutations in endometrial glands.

Major Comment #6.

For the estimates of the timings of CN-LOH and clonal expansions, please provide a measure of uncertainty, possibly as a confidence interval.

Response.

We estimated the 95% confidence intervals for the timings of CN-LOH and clonal expansions. We revised the methods and added the estimates in the main text.

Major Comment #7.

For subject 35, why were the anterior and posterior walls analyzed independently? The authors conclude independent FGFR2 mutated at a hotspot but this seems to require more explanation and justification. Why is the non-contiguous distribution of FGFR2 S252W not the result of the rhizome structure?

Response.

In subject 35, there were no glands sharing multiple mutations between the anterior and posterior walls, suggesting that mutant cell clones in the anterior and posterior walls were different due to their spatial separation. To depict this finding, the results from the anterior and posterior walls were shown in independent figures.

Based on the results of WES, glands in clusters 3, 8, 12, 13, 14 and 15 shared only the hotspot mutation of *FGFR2* (p.S252W) but the profiles for the other mutations were completely discordant. These clusters were located in spatially separated regions: Cluster 3 was distributed in the center of the anterior wall, cluster 12 was in the upper right of the posterior wall, and clusters 8, 13-15 were in the lower left of the posterior wall (Figure 4i). These three regions were far away. Please note that a rhizome structure is thought to extend only to neighboring area. There is a possibility that a rhizome structure expands to a

large part of the endometrium including the three regions. In such case, it is expected that glands derived from the same ancestral clone with the *FGFR2* mutation were present in regions connecting the three regions. However, we did not detect such glands. Therefore, it is not likely that the *FGFR2* mutation initially occurred in a single ancestral clone and the ancestral clone expanded to these three separated regions. It seems to be reasonable that at least three mutational events of the *FGFR2* mutation occurred independently in the three separated regions of the endometrium and the mutant clones diversified at each region by acquiring region-specific mutations.

According to the reviewer's comment, we re-structured the sentence as follows (pages 14-15):

Glands in six clusters had the same mutation of *FGFR2* (p.S252W). However, WES showed that the glands in different clusters did not share any mutations other than *FGFR2* (p.S252W) (Supplementary Fig. 5f). Additionally, glands in these six clusters were localized in spatially separated regions of the endometrium: Cluster 3 was distributed in the center of the anterior wall, cluster 12 was in the upper right of the posterior wall, and clusters 8, 13-15 were in the lower left of the posterior wall (Figure 4i). It is not likely that the *FGFR2* initially occurred in a single ancestral clone and the ancestral clone expanded these three separated regions as depicted by the phylogenetic tree (Fig. 4l), because glands with the *FGFR2* mutation were not detected in regions connecting the three regions. Therefore, a plausible explanation may be that at least three mutational events of the *FGFR2* mutation occurred independently in the three separated regions of the endometrium and the mutant clones diversified at each region by acquiring region-specific mutations.

Minor Comment #1.

This sentence is somewhat confusing: "We used endometrial glands whose coverages of the target region of at least 20 reads were greater than 0.70 for subsequent analyses."

Response.

According to the reviewer's comment, we re-structured the sentence as follows (page 33):

We used endometrial glands for subsequent analyses if more than 70% of the target bases were covered by at least 20 reads.

Minor Comment #2.

This sentence could be written more clearly: “We identified significant SBS signatures if the lower end of the 90% highest posterior density interval for the estimated exposure was greater than the threshold value (0.01).”

Response.

According to the reviewer’s comment, we re-structured the sentence as follows (page 35):

We estimated the highest posterior density (HPD) interval for each of the SBS signatures. If the 90% lower end of the HPD interval for a SBS signature was above the threshold (0.01, default value), we considered that the SBS signature was significantly active signature.

Minor Comment #3.

What form of multiple test correction was used for identifying significant positively selected genes?

Response.

In the revised version of our study, we used the Benjamini-Hochberg procedure to control the false discovery rate at 0.1 for the identification of positively selected genes. We added descriptions on this analysis in the Methods section (pages 38-39).

Additional Corrections.

Correction #1.

In Results, we corrected one character of amino acid of PIK3CA (p.Q545A) and PIK3CA (p.Q542V) to PIK3CA (p.E545A) and PIK3CA (p.E542V), respectively.

Correction #2.

In Discussion, we added the sentence of somatic mutations of endometrium-related diseases as follows (pages 25-26):

In the WES study for a patient with ovarian clear cell carcinoma with concurrent endometriosis, we showed that many somatic mutations including

cancer-associated gene mutations were shared among epithelium samples from uterine endometrium, endometriotic lesions distant from and adjacent to the carcinoma, and the carcinoma (Suda et al. Cancer Sci 2020).

Correction #3.

In Methods, we added the information of the average sequencing depth and the coverage of the target regions.

Correction #4.

In Figure legends, we added the definitions of the box-plot elements and scale bars (Figures 1d, 1g, 2b-e, Supplementary Figures 2a,b,d,e, 10b).

Correction #5.

In Figure 2, Supplementary Figures 2 and 3, p-values less than 0.01 were unified into exponential notation.

Correction #6.

In Figure 4a,d,c and legends, we added the information of the direction of uterus.

Correction #7.

In Figure 4l, we corrected the position of the square object of the gland No.46-1.

Correction #8.

In Figure 7e, we changed the color of the heat map.

REVIEWERS' COMMENTS

Reviewer #1 (Remarks to the Author):

The reviewer have addressed most of my concerns and I believe the manuscript has improved. In my opinion, analysing the merged data would have certainly helped with adding more novel findings to the current manuscript. However, it is understandable if this cannot be achieved in a timely manner. I have no further comments.

Reviewer #2 (Remarks to the Author):

I am satisfied by the response and changes provided by the authors.

Reviewer #3 (Remarks to the Author):

I thank the authors for their work and for addressing my concerns. I have the following minor comments:

1. The authors rewrote the following sentence and made it more confusing and less correct:

SNVs with low MAF (<0.1) in multiple samples, a characteristic of suspected false-positive mutation calls in genes such as TTN and mucin family genes probably due to misalignment of reads, were excluded.

Predicted mutations are more frequent in genes such as TTN because it is a larger gene, and this includes low frequency artifacts. If the low frequency mutations are common across multiple datasets including a panel of unrelated samples, that would be better evidence they are artifacts. Nevertheless, even though its possible the authors are excluding some real mutations with this filter, I dont think it would make a material difference to the findings in the paper and I suggest they just write:

SNVs with low MAF (<0.1) in multiple samples were excluded as putative false positives.

2. In supplementary 6f, the FGFR2 mutation is difficult to see in the heatmap, yet is important for the results.

We wish to express our strong appreciation to the editor and three reviewers for their valuable comments on our paper. As indicated in the responses below, we have taken all these comments and suggestions into considerations.

Response to the comments by Reviewer #1.

Remarks to the Author.

The reviewer have addressed most of my concerns and I believe the manuscript has improved. In my opinion, analysing the merged data would have certainly helped with adding more novel findings to the current manuscript. However, it is understandable if this cannot be achieved in a timely manner. I have no further comments.

Response.

We deeply appreciate the reviewer's positive evaluation on our manuscript.

Response to the comments by Reviewer #2.

Remarks to the Author.

I am satisfied by the response and changes provided by the authors.

Response.

We strongly appreciate the reviewer's positive evaluation on our manuscript.

Response to the comments by Reviewer #3.

Remarks to the Author.

I thank the authors for their work and for addressing my concerns. I have the following minor comments:

1. The authors rewrote the following sentence and made it more confusing and less correct:

SNVs with low MAF (<0.1) in multiple samples, a characteristic of suspected

false-positive mutation calls in genes such as TTN and mucin family genes probably due to misalignment of reads, were excluded.

Predicted mutations are more frequent in genes such as TTN because it is a larger gene, and this includes low frequency artifacts. If the low frequency mutations are common across multiple datasets including a panel of unrelated samples, that would be better evidence they are artifacts. Nevertheless, even though its possible the authors are excluding some real mutations with this filter, I don't think it would make a material difference to the findings in the paper and I suggest they just write:

SNVs with low MAF (<0.1) in multiple samples were excluded as putative false positives.

Response.

We deeply appreciate that reviewer #3 provided valuable comments to improve our manuscript. We agree with the reviewer's comment and have reflected this comment in Methods section.

2. In supplementary 6f, the FGFR2 mutation is difficult to see in the heatmap, yet is important for the results.

Response.

Thank you for your valuable suggestion. According to the reviewer's comment, we expanded the heatmap vertically and made it easier to see the *FGFR2* mutation in Supplementary Figure 6f. The layout of Supplementary Figure 6g-i was changed accordingly.

Additional Correction

In Figure 2c and Supplementary Figure 2a and 3c, we excluded the labels 0.5 and 1.5 from the x-axis since 'Parity' was an integral number.